# DIMENSION-FREE BOUNDS FOR LOW-PRECISION TRAINING

## ABSTRACT

Low-precision training is a promising way of decreasing the time and energy cost of training machine learning models. Previous work has analyzed low-precision training algorithms, such as low-precision stochastic gradient descent, and derived theoretical bounds on their convergence rates. These bounds tend to depend on the dimension of the model $d$ in that the number of bits needed to achieve a particular error bound increases as $d$ increases. This is undesirable because a motivating application for low-precision training is large-scale models, such as deep learning, where $d$ can be huge. In this paper, we prove dimension-independent bounds for low-precision training algorithms that use fixed-point arithmetic, which lets us better understand what affects the convergence of these algorithms as parameters scale. Our methods also generalize naturally to let us prove new convergence bounds on low-precision training with other quantization schemes, such as low-precision floating-point computation and logarithmic quantization.

## 1 INTRODUCTION

As machine learning models continue to scale to target larger problems on bigger data, the task of training these models quickly and efficiently becomes an ever-more-important problem. One promising technique for doing this is *low-precision computation*, which replaces the 32-bit or 64-bit floating point numbers that are usually used in ML computations with smaller numbers, often 8-bit or 16-bit fixed point numbers. Low-precision computation is a broadly applicable technique that has received a lot of attention, especially for deep learning, and specialized hardware accelerators have been developed to support it (Jouppi et al., 2017; Burger, 2017; Caulfield et al., 2017).

A major application for low-precision computation is the training of ML models using empirical risk minimization. This training is usually done using *stochastic gradient descent* (SGD), and most research in low-precision training has focused on low-precision versions of SGD. While most of this work is empirical (Wu et al., 2018; Das et al., 2018; Zhu et al., 2016; Köster et al., 2017; Lee et al., 2017; Hubara et al., 2016; Rastegari et al., 2016; Zhou et al., 2016; Gupta et al., 2015; Courbariaux et al., 2014; 2015; De Sa et al., 2017), significant research has also been done in the theoretical analysis of low-precision training. This theoretical work has succeeded in proving bounds on the convergence rate of low-precision SGD and related low-precision methods in various settings, including for convex (De Sa et al., 2018; Zhang et al., 2017) and non-convex objectives (De Sa et al., 2015; Li et al., 2017; Alistarh et al., 2017). One common characteristic of these results is that the bounds tend to depend on the dimension $d$ of the model being learned (equivalently, $d$ is the number of parameters). For example, (Li et al., 2017) gives the convergence bound

$$\mathbf{E}\left[f(\bar{w}_T) - f(w^*)\right] \leq \frac{(1 + \log(T+1))\sigma_{\max}^2}{2\mu T} + \frac{\sigma_{\max}\delta\sqrt{d}}{2}, \tag{1}$$

where the objective $f$ is strongly convex with parameter $\mu$, low-precision SGD outputs $\bar{w}_T$ after $T$ iterations, $w^*$ is the true global minimizer of the objective, $\sigma_{\max}^2$ is an upper bound on the second moment of the stochastic gradient samples $\mathbf{E}[\|\tilde{f}(w)\|_2^2] \leq \sigma_{\max}^2$, and $\delta$ is the quantization step, the difference between adjacent numbers in the low-precision format. Notice that, as $T \to \infty$, this bound shows convergence down to a level of error that increases with the dimension $d$. Equivalently, in order to achieve the same level of error as $d$ increases, we would need to use more bits of quantization to make $\delta$ smaller. Similar dimension-dependent results, where either the error or the number of bits needed increases with $d$, can also be seen in other work on low-precision training algorithms (Alistarh et al., 2017; Zhang et al., 2017; De Sa et al., 2018). This dependence on $d$ is unsatisfying because the motivation for low-precision training is to tackle large-scale problems on big data, where $d$ can range up to $10^8$ or more for commonly used models (Simonyan and Zisserman,

Table 1: Summary of our dimension-free results compared with prior work. The values report the number of bits needed, according to the theoretical bound, for the LP-SGD (Li et al., 2017) algorithm to achieve an expected objective gap ($f(w) - f(w^*)$) of $\epsilon$ when we let step size $\alpha \to 0$, epoch length $T \to \infty$. Here we let $R$ denote the radius of the range of numbers representable in the low-precision format and assume $\|w^*\|_2 = \Theta(R)$. The rest of the parameters can be found in the assumptions to be introduced later.

| NUMBER OF BITS NEEDED FOR $\mathbf{E}\left[f(w) - f(w^*)\right] \leq \epsilon$ | |
| --- | --- |
| PRIOR DIMENSION-DEPENDENT BOUND | $\log_2 \mathcal{O}\big(R\sigma_{\max}\sqrt{d}/\varepsilon\big)$ |
| OUR DIMENSION-FREE BOUND | $\log_2 \mathcal{O}\big(R\sigma_1/\varepsilon\big)$ |
| DIMENSION-FREE WITH LOGARITHMIC QUANTIZATION | $\log_2 \mathcal{O}\big((R\sigma/\varepsilon) \cdot \log\big(1 + \sigma_1/\sigma\big)\big)$ |

2014). For example, to compensate for a factor of $d = 10^8$ in (1), we could add bits to decrease the quantization step $\delta$ by a factor of $\sqrt{d}$, but this would require adding $\log_2(10^4) \approx 13$ bits, which is significant compared to the 8 or 16 bits that are commonly used in low-precision training.

In this paper, we address this problem by proving *dimension-free bounds* on the convergence of LP-SGD Li et al. (2017). Our main technique for doing so is a tight dimension-independent bound on the expected quantization error of the low-precision stochastic gradients in terms of the $\ell_1$-norm. Our results are summarized in Table 1, and we make the following contributions:

- We describe conditions under which we can prove a dimension-free bound on the convergence of SGD with fixed-point, quantized iterates on strongly convex problems.

- We study non-linear quantization schemes, in which the representable low-precision numbers are distributed non-uniformly. We prove dimension-free convergence bounds for SGD using logarithmic quantization (Lee et al., 2017), and we show that using logarithmic quantization can reduce the number of bits needed for LP-SPG to provably converge.

- We study quantization using low-precision floating-point numbers, and we present theoretical analyis that suggests how to assign a given number of bits to exponent and mantissa to optimize the accuracy of training algorithms. We validate our results experimentally.

## 2 RELATED WORK

Motivated by the practical implications of faster machine learning, much work has been done on low-precision training. This work can be roughly divided into two groups. The first focuses on training deep models with low-precision weights, to be later used for faster inference. For some applications, methods of this type have achieved good results with very low-precision models: for example, binarized (Courbariaux et al., 2015; Hubara et al., 2016; Rastegari et al., 2016) and ternary networks (Zhu et al., 2016) have been observed to be effective (although as is usual for deep learning they lack theoretical convergence results). However, these approaches are still typically trained with full-precision iterates: the goal is faster inference, not faster training (although faster training is often achieved as a bonus side-effect).

A second line of work on low-precision training, which is applied to both DNN training and non-deep-learning tasks, focuses on making various aspects of SGD low-precision, while still trying to solve the same optimization problem as the full-precision version. The most common way to do this is to make the iterates of SGD (the $w_t$ in the SGD update step $w_{t+1} = w_t - \alpha_t \nabla f_t(w_t)$) stored and computed in low-precision arithmetic (Courbariaux et al., 2014; Gupta et al., 2015; De Sa et al., 2018; 2015; Li et al., 2017). This is the setting we will focus on most in this paper, because it has substantial theoretical prior work which exhibits the dimension-dependence we set out to study (Li et al., 2017; Zhang et al., 2017; Alistarh et al., 2017; De Sa et al., 2018). The only paper we found with a bound that was not dimension-dependent was De Sa et al. (2015), but in that paper the authors required that the gradient samples be 1-sparse (have only one nonzero entry), which is not a realistic assumption for most ML training tasks. In addition to quantizing the iterates, other work has studied quantizing the training set (Zhang et al., 2017) and numbers used to communicate among parallel workers (Alistarh et al., 2017). We expect that our results on dimension-free bounds will be complementary with these existing theoretical approaches, and we hope that they can help to explain the success of the exciting empirical work in this area.

## 3 DIMENSION-FREE BOUNDS FOR SGD

In this section, we analyze the performance of stochastic gradient descent (SGD) using low-precision training. Though there are numerous variants of this algorithm, SGD remains the *de facto* algorithm

used most for machine learning. We will start by describing SGD and how it can be made low-precision. Suppose we are trying to solve the problem

$$\text{minimize: } f(w) = \frac{1}{n} \sum_{i=1}^{n} \tilde{f}_i(w) \qquad \text{over: } w \in \mathbb{R}^d. \tag{2}$$

SGD solves this problem iteratively by repeatedly running the update step

$$w_{t+1} = w_t - \alpha \nabla \tilde{f}_{i_t}(w_t) \tag{3}$$

where $\alpha$ is the *step size*[1] or learning rate, and $i_t$ is the index of a component function chosen randomly and uniformly at each iteration from $\{1, \ldots, n\}$. To make this algorithm low-precision, we *quantize the iterates* (the vectors $w_t$) and store them in a low-precision format. The standard format to use lets us represent numbers in a set

$$\text{dom}(\delta, b) = \{-\delta \cdot 2^{b-1}, \cdots, -\delta, 0, \delta, \cdots, \delta \cdot (2^{b-1} - 1)\}$$

with $\delta > 0$ being the *quantization gap*, the distance between adjacent representable numbers, and $b \in \mathbb{N}$ being the number of bits we use (De Sa et al., 2018). Usually, $\delta$ is a power of 2, and this scheme is called *fixed-point arithmetic*. It is straightforward to encode numbers in this set as $b$-bit signed integers, by just multiplying or dividing by $\delta$ to convert to or from the encoded format—and we can even do many arithmetic computations on these numbers directly as integers. This is sometimes called *linear quantization* because the representable points are distributed uniformly throughout their range. However, as the gradient samples will produce numbers outside this set during iteration, we need some way to map these numbers to the set of numbers that we can represent. The standard way to do this is with a *quantization function $Q(x) : \mathbb{R} \to \text{dom}(\delta, b)$*. While many quantization functions have been proposed, the one typically used in theoretical analysis (which we will continue to use here) is *randomized rounding*. Randomized rounding, also known as unbiased rounding or stochastic rounding, rounds up or down at random such that $\mathbf{E}[Q(x)] = x$ whenever $x$ is within the range of representable numbers (i.e. when $-\delta \cdot 2^{b-1} \leq x \leq \delta \cdot (2^{b-1} - 1)$). When $x$ is outside that range, we quantize it to the closest representable point. When we apply $Q_{(\delta,b)}$ to a vector argument, it quantizes each of its components independently.

Using this quantization function, we can write the update step for low-precision SGD (LP-SGD), which is a simple quantization of (3),

$$w_{t+1} = Q\left(w_t - \alpha \nabla \tilde{f}_{i_t}(w_t)\right) \tag{4}$$

As mentioned before, one common feature of prior bounds on the convergence of LP-SGD is that they depend on the number of dimensions $d$, whereas bounds on full precision SGD under the same conditions do not do so. This difference is due to the fact that, when we use a quantization function $Q$ to quantize a number $w$, it increases its variance by $\mathbf{E}\left[(Q(w) - w)^2\right] \leq \delta^2/4$. Observe that this inequality is tight since it holds as an equality when $w$ is in the middle of two quantization points, e.g. $w = \delta/2$, as illustrated in Figure 1(a). When quantizing a vector $w \in \mathbb{R}^d$, the squared error can be increased by

$$\mathbf{E}\left[\|Q(w) - w\|_2^2\right] = \sum_{k=1}^{d} \mathbf{E}\left[(Q(w_k) - w_k)^2\right] \leq \frac{\delta^2 d}{4}, \tag{5}$$

and this bound is again tight. This variance inequality is the source of the $d$ term in analyses of LP-SGD, and the tightness of the bound leads to the natural belief that the $d$ term is inherent, and that low-precision results are inevitably dimension-dependent.

However, we propose that if we can instead bound the variance in (5) with some properties of the problem itself that do not change as $d$ changes, we can achieve a result that is dimension-independent. One way to do this is to look at the variance graphically. Figure 3 plots the quantization error as a function of $w$ along with the bound in (5). Notice that the squared error looks like a series of parabolas, and the bound in (5) is tight at the top of those parabolas, but loose elsewhere. Instead, suppose we want to do the opposite and produce a bound that is tight when the error is zero (at points in $\text{dom}(\delta, b)$). To do this, we observe that $\mathbf{E}\left[(Q(w) - w)^2\right] \leq \delta|w - z|$ for any $z \in \text{dom}(\delta, b)$. This bound is also tight when $z$ is adjacent to $w$, and we plot it in Figure 3 as well. The natural vector analog of this is

$$\mathbf{E}\left[\|Q(w) - w\|_2^2\right] \leq \sum_{k=1}^{d} \delta|w_k - z_k| = \delta \|w - z\|_1, \ \forall z \in \text{dom}(\delta, b)^d \tag{6}$$

---

[1]Usually in SGD the step size is decreased over time, but here for simplicity we consider a constant learning rate schedule.

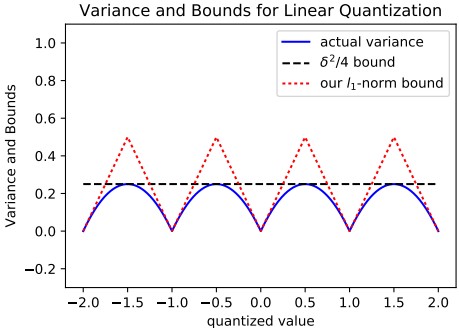

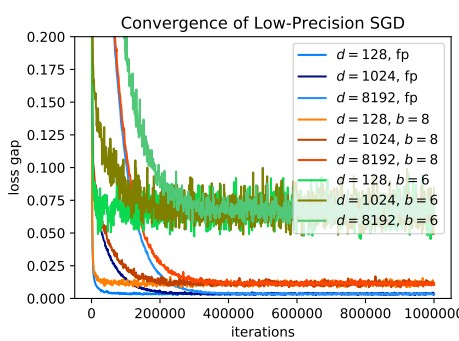

(a) The squared quantization error $\mathbf{E}\left[(Q(w) - w)^2\right]$ and two possible tight upper bounds in the one-dimensional case.

(b) The convergence of training loss gap $f(w) - f(w^*)$ as we train low-precision SGD, showing the effect of different model sizes and precision.

Figure 1: (a) Quantization error bounds; (b) Convergence of full-precision (fp) SGD and LP-SGD.

where $\|\cdot\|_1$ denotes the L1 norm. This is a dimension-independent bound we can use to replace (5) to bound the convergence of LP-SGD and other algorithms. However, this replacement is nontrivial as our bound is now non-constant: it depends on $w$, which is a variable updated each iteration. Also, in order to bound this new L1 norm term, we will need some new assumptions about the problem. Next, we will state these assumptions, along with the standard assumptions used in the analysis of SGD for convex objectives, and then we will use them to present our dimension-free bound on the convergence of SGD.

**Assumption 1.** *All the loss functions $\tilde{f}_i$ are differentiable, and their gradients are L-Lipschitz continuous in the sense of 2-norm, that is,*

$$\forall i \in \{1, 2, \cdots, n\}, \quad \forall x, y \in \mathbb{R}^d, \quad \|\nabla \tilde{f}_i(x) - \nabla \tilde{f}_i(y)\|_2 \leq L \|x - y\|_2$$

**Assumption 2.** *All the gradients of the loss functions $\tilde{f}_i$ are $L_1$-Lipschitz continuous in the sense of 1-norm to 2-norm, that is,*

$$\forall i \in \{1, 2, \cdots, n\}, \quad \forall x, y \in \mathbb{R}^d, \quad \|\nabla \tilde{f}_i(x) - \nabla \tilde{f}_i(y)\|_1 \leq L_1 \|x - y\|_2$$

These two assumptions are simply expressing of Lipschitz continuity in different norms. Assumption 1 is a standard assumption in the analysis of SGD on convex objectives, and has been applied in the low-precision case as well in prior work (De Sa et al., 2018). Assumption 2 is analogous to 1, except we are bounding the L1 norm instead of the L2 norm. This holds naturally (with a reasonable value of $L_1$) for many problems, in particular problems for which the gradient samples are sparse.

**Assumption 3.** *The gradient of the total loss function $f$ is $\mu$-strongly convex for some $\mu > 0$:*

$$\forall w, v, \quad f(w) - f(v) - \frac{\mu}{2} \|w - v\|_2^2 \geq (w - v)^T \nabla f(v)$$

Assumption 3 is a standard assumption that bounds the curvature of the loss function $f$, and is satisfied for many classes of convex objectives. For example, any convex loss with L2 regularization will always be strongly convex. When an objective is strongly convex and Lipschitz continuous, it is standard to say it has *condition number* $\kappa = L/\mu$, and here we extend this to say it has *L1 condition number* $\kappa_1 = L_1/\mu$.

**Assumption 4.** *The gradient of each loss function is bounded by some constant $\sigma$ near the optimal point in the sense of $l_1$ and $l_2$ norm, that is,*

$$\mathbf{E}\left[\left\|\nabla \tilde{f}_i(w^*)\right\|_2^2\right] \leq \sigma^2, \qquad \mathbf{E}\left[\left\|\nabla \tilde{f}_i(w^*)\right\|_1\right] \leq \sigma_1$$

This assumption constrains the gradient for each loss function at the optimal point. We know $\nabla f(w^*) = \frac{1}{n} \sum_i \tilde{\nabla} f_i(w^*) = 0$, so it is intuitive that each $\nabla \tilde{f}_i(w^*)$ can be bounded by some value. Therefore this is a natural assumption to make and it has been used in a lot of other work in this area. Note that this assumption only needs to hold under the expectation over all $\tilde{f}_i$. With these assumptions, we proved the following theorem for low-precision SGD:

**Theorem 1.** *Suppose that we run LP-SGD on an objective that satisfies Assumptions 1–4, and with step size $\alpha < 1/(2\kappa^2\mu)$. After $T$ LP-SGD update steps (4), select $\bar{w}_T$ uniformly at random from $\{w_0, w_1, \ldots, w_{T-1}\}$. Then, the expected objective gap of $\bar{w}_T$ is bounded by*

$$\mathbf{E}\left[f(\bar{w}_T) - f(w^*)\right] \leq \frac{1}{2\alpha T} \|w_0 - w^*\|_2^2 + \frac{\alpha \sigma^2 + \delta \sigma_1}{2} + \frac{\delta^2 \kappa_1^2 \mu}{4}$$

This theorem shows a bound of the expected distance between the result we get at $K$-th iteration and the optimal value. By choosing an appropriate step size we can achieve convergence at a $1/T$ rate, while the limit we converge to is only dependent on dimension-free factors. Meanwhile, as mentioned in the first section, previous work gives a dimension-dependent bound (1) for the problem, which also converges at a $1/T$ rate.[2] Therefore our result guarantees a dimension-independent convergence limit without weakening the convergence rate.

It is important to note that, because the dimension-dependent bound in (5) was tight, we should not expect our new result to improve upon the previous theory in all cases. In the worst case, $\kappa_1 = \sqrt{d} \cdot \kappa$ and similarly $\sigma_1 = \sqrt{d} \cdot \sigma$; this follows from the fact that for vectors in $\mathbb{R}^d$, the norms are related by the inequality $\|x\|_1 \leq \sqrt{d} \cdot \|x\|_2$. Substituting this into our result produces a dimension-dependent bound again. This illustrates the importance of introducing the new parameters $\kappa_1$ and $\sigma_1$ and requiring that they be bounded; if we could not express our bound in terms of these parameters, the best we could do here is recover a dimension-dependent bound.

**Experiments** Next, we validate our theoretical results experimentally. To do this, we analyzed how the size of the noise floor of convergence of SGD and LP-SGD varies as the dimension is changed for a class of synthetic problems. Importantly, we needed to pick a class of problems for which the parameters $L$, $L_1$, $\mu$, $\sigma$, and $\sigma_1$, did not change as we changed the dimension $d$. To do this, we chose a class of synthetic linear regression models with loss components sampled independently and identically as

$$\tilde{f}_i(w) = \frac{1}{2}(\tilde{x}^T w - \tilde{y})^2$$

where $\tilde{x}$ is a sparse vector sampled to have $s$ nonzero entries each of which is sampled uniformly from $\{-1, 1\}$, and $\tilde{y}$ is sampled from $\mathcal{N}(\tilde{x}^T w^*, \beta^2)$ for some variance parameter $\beta$. Importantly, the nonzero entries of $\tilde{x}$ were chosen non-uniformly such that $\Pr[\tilde{x}_i \neq 0] = p_i$ for some probabilities $p_i$ which decrease as $i$ increases; this lets us ensure that $\mu$ remains constant as $d$ is increased. For simplicity, we sampled a fresh loss component of this form at each SGD iteration, which is sometimes called the *online* setting. It is straightforward to derive that for this problem

$$\mu = p_d \qquad L = s \qquad L_1 = s\sqrt{s} \qquad \sigma^2 = \beta^2 s \qquad \sigma_1 = \sqrt{2s/\pi}\sigma.$$

We set $\alpha = 0.01$, $\beta = 0.2$, $p_1 = 0.9$, $p_d = 0.001$, and $s = 16$, we chose each entry of $w^*$ uniformly from $[-1/2, 1/2]$, and we set $\delta$ such that the low-precision numbers would range from $-1$ to $1$. Figure 1(b) shows the convergence of SGD and LP-SGD as the dimension $d$ is changed, for both 8-bit and 6-bit quantization. Notice that while changing $d$ has an effect on the initial convergence rate for both SGD and LP-SGD, it has no effect on the noise ball size, the eventual loss gap that the algorithm converges to. Figure 2(a) measures this noise ball size more explicitly as the dimension is changed: it reports the loss gap averaged across the second half of the iterates. Notice that as the dimension $d$ is changed, the average loss gap is almost unchanged, even for very low-precision methods for which the precision *does* significantly affect the size of the noise ball. This validates our dimension-free bounds, and shows that they can describe the actual dependence on $d$ in at least one case.

Figure 2(b) validates our results in the opposite way: it looks at how this gap changes as our new parameters $\sigma_1$ and $L_1$ change while $d$, $\mu$, and $\sigma$ are kept fixed. To do this, we fixed $d = 1024$ and changed $s$ across a range, setting $\beta = 0.8/\sqrt{s}$, which keeps $\sigma^2$ constant as $s$ is changed: this has the effect of changing $\sigma_1$ (and, as a side effect, $L_1$ and $L$). We can see from figure 2(b) that changing $\sigma_1$ in this way has a much greater effect on LP-SGD than it does on SGD. This validates our theoretical results, and suggests that $\sigma_1$ and $L_1$ can effectively determine the effect of low-precision compute on SGD.

## 4 NON-LINEAR QUANTIZATION

Up till now, most theoretical work in the area of low-precision machine learning has been on linear quantization, where the distance between adjacent quantization points is a constant value $\delta$. Another option is *non-linear quantization* (NLQ), in which we quantize to a set of points that are non-uniformly distributed. This approach has been shown to be effective for accelerating deep learning

---

[2]Previous work (1) used a decaying step size while ours uses a constant step size to achieve a better result.

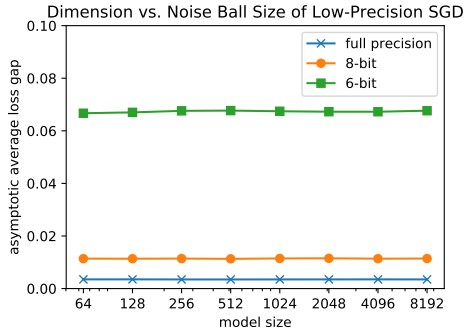
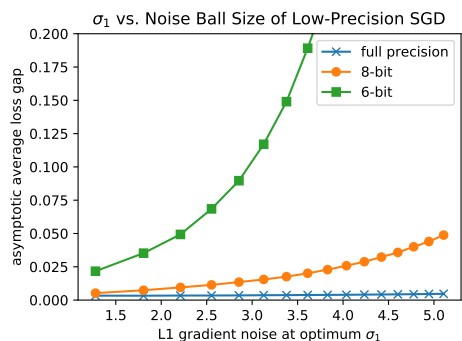

(a) The size of the noise ball is not significantly affected by model size $d$.

(b) The size of the noise ball does depend on $\sigma_1$, especially when precision is low.

Figure 2: Plots of the asymptotic loss gap from Figure 1(b) as a function of model size $d$ and $\sigma_1$.

in some settings (Lee et al., 2017). In general, we can quantize to a set of points

$$D = \{-q_n, \cdots, -q_1, q_0, q_1, \cdots, q_{n-1}\},$$

and, just like with linear quantization, we can still use a quantization function $Q(w)$ with randomized rounding that rounds up or down to a number in $D$ in such a way that $\mathbf{E}[Q(w)] = w$ for $w \in [-q_n, q_{n-1}]$. When we consider the quantization variance here, the natural dimension-dependent bound would be

$$\mathbf{E}\left[\|Q(w) - w\|_2^2\right] \le \frac{d}{4}\max_i(q_i - q_{i-1})^2.$$

This is still a tight bound since it holds with equality for a number in the middle of two most distant quantization points. However, when applied in the analysis of LP-SGD, this bound induces poor performance and often under-represents the actual result.

Here we discuss a specific NLQ method and use it to introduce a tight bound on the quantization variance. This method has been previously studied as *logarithmic quantization* or $\mu-$law quantization, and is defined recursively by

$$q_0 = 0, \qquad q_{i+1} - q_i = \delta + \zeta q_i \tag{7}$$

where $\delta > 0$ and $\zeta > 0$ are fixed parameters. Note that this includes linear quantization as a special case by setting $\zeta = 0$. It turns out that we can prove a tight dimension-independent bound on the quantization variance of this scheme. First, we introduce the following definition.

**Definition 1.** *An unbiased quantization function $Q$ satisfies the dimension-free variance bound with parameters $\delta$, $\zeta$, and $\eta$ if for all $w \in [-q_n, q_{n-1}]$ and all $z \in D$,*

$$\mathbf{E}\left[\|Q(w) - w\|_2^2\right] \le \delta \|w - z\|_1 + \zeta \|z\|_2 \cdot \|w - z\|_2 + \eta \|w - z\|_2^2.$$

We can prove that our logarithmic quantization scheme satisfies this bound.

**Lemma 1.** *The logarithmic quantization scheme (7) satisfies the dimension-free variance bound with parameters $\delta$, $\zeta$, and $\eta = \frac{\zeta^2}{4(\zeta+1)} < \frac{\zeta}{4}$.*

Notice that this bound becomes identical to the linear quantization bound (6) when $\zeta = 0$, so this result is a strict generalization of our results from the linear quantization case. With this setup, we can apply NLQ to the low-precision training algorithms we have studied earlier in this paper.

**Theorem 2.** *Suppose that we run LP-SGD on an objective that satisfies Assumptions 1–4, and using a quantization scheme that satisfies the dimension-free variance bound. If $\zeta < \frac{1}{\kappa}$, then*

$$\mathbf{E}\left[(f(\bar{w}_T) - f(w^*))\right] \le \frac{\|w_0 - w^*\|_2^2}{2\alpha T} + \frac{(1+\eta)\alpha\sigma^2 + \delta\sigma_1 + \zeta\sigma\|w^*\|_2}{2} + \frac{(\delta L_1 + \zeta L\|w^*\|_2 + \zeta\sigma)^2}{4\mu}$$

This theorem is consistent with Theorem 1 in that, if we set $\zeta = \eta = 0$, which makes logarithmic quantization linear, they would have an identical result. If we fix the representable range $R$ (the largest-magnitude values representable in the low-precision format) and choose our quantization

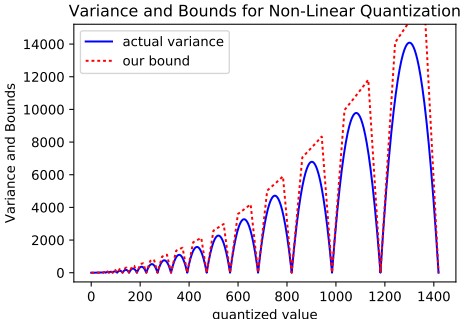 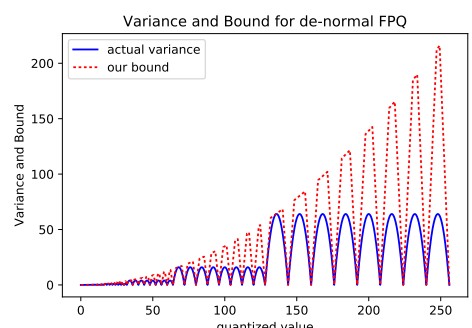

(a) variance and bound for $\mu$–law quantization  (b) variance and bound for FPQ

Figure 3: A figure showing the actual quantization variance $\mathbf{E}\left[\|Q(w) - w\|_2^2\right]$ and the tight upper bound that we introduced in one dimension. Similarly to 1(a) we plot this bound when taking the minimum over all possible $z$.

parameters optimally, we get the result that the number of bits we need to achieve objective gap $\epsilon$ is $\log_2 \mathcal{O}\big((R\sigma/\varepsilon)\cdot\log\big(1+\sigma_1/\sigma\big)\big)$. This bound is notable because even in the worst case where we do not have a bound on $\sigma_1$ and must use $\sigma_1 \le \sqrt{d}\cdot\sigma$, this bound gives us $\log_2 \mathcal{O}\big((R\sigma/\varepsilon)\cdot\log\big(1+\sqrt{d}\big)\big)$. That is, while a dimension-dependent factor still remains, it is now "hidden" within a log term. This greatly decreases the effect of the dimension, and suggests that NLQ may be a promising technique to use for low-precision training at scale. Also note that, although this bound holds only when we set $\zeta < \frac{1}{\kappa} = \frac{\mu}{L}$, which to some extent limits the acceleration of the strides in logarithmic quantization, the bound $\frac{\mu}{L}$ is independent of $\sigma$ and $\sigma_1$, thus this effect of "pushing " $\sigma_1$ into a log term is independent of the setting of $\zeta$.

**Floating point.** Next, we look at another type of non-linear quantization that is of great practical use: *floating-point quantization (FPQ)*. Here, the quantization points are simply floating-point numbers with some fixed number of exponential bits $b_e$ and mantissa bits $b_m$. Floating-point numbers are represented in the form

$$(-1)^{\text{sign bit}} \cdot 2^{\text{exponent}-\text{bias}} \cdot \big(1.m_1m_2m_3\ldots m_{b_m}\big) \tag{8}$$

where "exponent" is a $b_e$-bit unsigned number, the $m_i$ are the $b_m$ bits of the mantissa, and "bias" is a term that sets the range of the representable numbers by determining the range of the exponent. In standard floating point numbers, the exponent ranges from $[-2^{b_e-1}+2, 2^{b_e-1}-1]$, which corresponds to a bias of $2^{b_e-1}-1$. To make our results more general, we also consider non-standard bias by defining a *scaling factor* $s = 2^{-(\text{bias}-\text{standard bias})}$; the standard bias setting corresponds to $s = 1$. We also consider the case of *denormal* floating point numbers, which tries to address underflow by replacing the 1 in (8) with a 0 for the smallest exponent value. Under these conditions, we can prove that floating-point quantization satisfies the bound in Definition 1.

**Lemma 2.** *The FPQ scheme using randomized rounding satisfies the dimension-free variance bound with parameters $\delta_{normal}, \zeta$, and $\eta$ for normal FPQ and $\delta_{denormal}, \zeta$, and $\eta$ for denormal FPQ where*

$$\delta_{normal} = \frac{4s}{2^{2^{b_e}}}, \qquad \delta_{denormal} = \frac{8s}{2^{2^{b_e}+b_m}}, \qquad \zeta = 2^{-b_m}, \qquad \eta = \frac{\zeta^2}{4(\zeta+1)}.$$

This bound can be immediately combined with Theorem 2 to produce dimension-independent bounds on the convergence rate of low-precision floating-point SGD. If we are given a fixed number of total bits $b = b_e + b_m$, we can minimize this upper bound on the objective gap to try to predict the best way to allocate our bits between the exponent and the mantissa. Unfortunately, there is no analytical expression for this optimal choice of $b_e$. To give a sense of the asymptotic behavior of this optimal allocation, we present upper and lower bounds on it.

**Theorem 3.** *When using FPQ without denormal numbers, given $b$ total bits, the optimal number of exponential bits $b_e$ such that the asymptotic upper bound on the objective gap given by Theorem 2 is minimized is in the interval between:*

$$\log_2\left[2\log_2\left(\frac{2(\ln 2)s\sigma_1}{\sigma\|w^*\|_2}\right) + 2b\right] \qquad \text{and} \qquad \log_2\left[2\log_2\left(\frac{2(\ln 2)sL_1}{L\|w^*\|_2+\sigma}\right) + 2b\right].$$

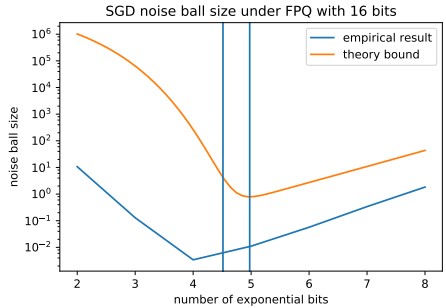
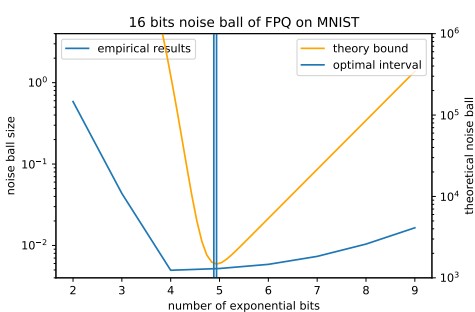

(a) Training noise ball size of SGD using 16 bits normal FPQ on synthetic data set

(b) Training noise ball size of SGD using 16 bits normal FPQ on MNIST

Figure 4: Plots of noise ball size vs. $b_e$ when running SGD with 16 bits FPQ on synthetic data set and MNIST. Note the use of two $y$-axes in Figure 4(b) to make the series fit in one figure.

**Theorem 4.** *When using denormal FPQ, given $b$ total bits, the optimal number of exponential bits $b_e$ such that the asymptotic upper bound on the objective gap, as $T \to \infty$ and $\alpha \to 0$, given by Theorem 2 is minimized is in the interval between:*

$$\log_2\left[1 - \frac{2}{\ln 2}W\left(\frac{e\sigma\|w^*\|_2}{8s\sigma_1}\right)\right] \qquad and \qquad \log_2\left[1 - \frac{2}{\ln 2}W\left(\frac{e(L\|w^*\|_2+\sigma)}{8sL_1}\right)\right]$$

*where $e$ denotes the base of the natural logarithm and $W$ stands for the* Lambert W *function. In cases where neither of these two values exists, the noise ball size increases as $b_e$, thus $b_e = 2$ would be the optimal setting, which is equivalent to linear quantization.*

These theorems give us an idea of where the optimal setting of $b_e$ lies such that the theoretical asymptotic error is minimized. When using normal FPQ, this optimal assignment of $b_e$ is $\mathcal{O}(\log(b))$, and for denormal FPQ the result is independent of $b$. This suggests that once the total number of bits grows past a threshold, we should assign most of or all the extra bits to the mantissa.

**Experiments**  For FPQ, we ran experiments on two different data sets. First, we ran LP-SGD on the same synthetic data set that we used for linear regression. Here we used normal FPQ with 20 bits in total, and we get the result in Figure 4(a). In this diagram, we plotted the empirical noise ball size, its theoretical upper bound, and the optimal interval for $b_e$ as Theorem 3 predicts. As the figure shows, our theorem accurately predicts the optimal setting of exponential bits, which is 5 in this case, to minimize both the theoretical upper bound and the actual empirical result of the noise ball size, despite the theoretical upper bound being loose.

Second, we ran LP-SGD on the MNIST dataset (Deng, 2012). To set up the experiment, we normalized the MNIST data to be in $[0, 1]$ by dividing by 255, then subtracted out the mean for each features. We ran multiclass logistic regression using an L2 regularization constant of $10^{-4}$ and a step size of $\alpha = 10^{-4}$, running for 500 total epochs (passes through the dataset) to be sure we converged. For this task, our (measured) problem parameters were $L = 37.41$, $L_1 = 685.27$, $\sigma = 2.38$, $\sigma_1 = 29.11$, and $d = 784$. In Figure 4(b), we plotted the observed loss gap, averaged across the last ten epochs, for LP-SGD using various 16-bit floating point formats. We also plot our theoretical bound on the loss gap, and the predicted optimal number of exponential bits to use based on that bound. Our results show that even though our bound is very loose for this task, it still predicts the right number of bits to use with reasonable accuracy. This experiment also validates the use of IEEE standard half-precision floating-point numbers, which have 5 exponential bits, for this sort of task.

## 5  CONCLUSION

In this paper, we present dimension-independent bounds on the convergence of SGD when applied to low-precision training. We point out the conditions under which such bounds hold. We further extend our results to non-linear methods of quantization: logarithmic quantization and floating point quantization. We analyze the performance of SGD under logarithmic quantization and demonstrate that NLQ is a promising method for reducing the number of bits required in low-precision training. We also presented ways in which our theory could be used to suggest how to allocate bits between exponent and mantissa when FPQ is used. We hope that our work will encourage further investigation of non-linear quantization techniques.

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

# A    ALGORITHM

In our work, we presented dimension-free bounds on the performance of low-precision SGD, here we present the algorithm in detail.

---

**Algorithm 1** LP-SGD: Low-Precision Stochastic Gradient Descent

---

   **given:** $n$ loss functions $\tilde{f}_i$, number of epochs $T$, step size $\alpha$, and initial iterate $w_0$.
   **given:** low-precision quantization function $Q$.
   **for** $t = 0$ **to** $T - 1$ **do**
       **sample** $i_t$ uniformly from $\{1, 2, \cdots, n\}$,
       **quantize** $w_{t+1} \leftarrow Q\left(w_t - \alpha \nabla \tilde{f}_{i_t}(w_t)\right)$
   **end for**
   **return** $w_T$

---

# B    PROOF FOR RESULTS IN TABLE 1

As mentioned in the caption of Table 1, here only we consider the convergence limit, that is, we assume $\alpha \to 0, T \to \infty$, and we compute the minimum number of bits $b$ we would require in order for the limit to be less than some small positive $\varepsilon$. Meanwhile, we denote the radius of the representable range by $R$ and we assume $R = \|w^*\|_2$ without loss of generality, as this is the worst case for all our bounds that depend on $\|w^*\|_2$. Then in linear quantization, we have:

$$q_{2^{b-1}-1} = \delta \cdot \left(2^{b-1} - 1\right) \geq R$$

and in non-linear quantization, we need:

$$q_{2^{b-1}-1} = \frac{\delta}{\zeta}\left((1+\zeta)^{\left(2^{b-1}-1\right)} - 1\right) \geq R \tag{9}$$

In the following proof we'll take the equality for these two inequalities.

## B.1    LP-SGD IN PREVIOUS WORK

In previous work Li et al. (2017), we have

$$f(\bar{w}_T) - f(w^*) \leq \frac{(1 + \log(T+1))G^2}{2\mu T} + \frac{G\delta\sqrt{d}}{2}$$

here we re-denote $G$ as $\sigma_{\max}$ for concordance with our result. Here $\sigma_{\max}^2$ is an upper bound on the second moment of the stochastic gradient samples $\mathbf{E}\left[\|\tilde{f}(w)\|_2^2\right] \leq \sigma_{\max}^2$. Substitute $\delta$ with $\frac{R}{2^{b-1}-1}$ and set the limit (as $\alpha \to 0$ and $T \to \infty$) to be $\leq \varepsilon$, and notice that $2^{b-1} - 1 > 2^{b-2}$, then we have:

$$\frac{\sigma_{\max}R\sqrt{d}}{2\left(2^{b-1}-1\right)} = \mathcal{O}\left(\varepsilon\right) \;\Rightarrow\; b \leq \log_2\left(\frac{\sigma_{\max}R\sqrt{d}}{\varepsilon}\right) + 1 = \log_2 \mathcal{O}\left(\frac{\sigma_{\max}R\sqrt{d}}{\varepsilon}\right)$$

## B.2    LP-SGD IN OUR WORK

In Theorem 1, we know that

$$\mathbf{E}\left[f(\tilde{w}) - f(w^*)\right] \leq \frac{1}{2\alpha T}\|w_0 - w^*\|_2^2 + \frac{\alpha\sigma^2 + \delta\sigma_1}{2} + \frac{\delta^2\kappa_1^2\mu}{4}$$

Set the limit (as $\alpha \to 0$ and $T \to \infty$) to be $\leq \varepsilon$, then we need:

$$\frac{\delta\sigma_1}{2} = \mathcal{O}\left(\varepsilon\right), \quad \frac{\delta^2\kappa_1^2\mu}{4} = \mathcal{O}\left(\varepsilon\right).$$

Then for sufficiently small $\varepsilon$, more explicitly, $\varepsilon$ that satisfies $\frac{L_1^2}{\mu\sigma_1^2}\mathcal{O}(\varepsilon) < 1$, setting

$$\delta = \mathcal{O}\left(\frac{\varepsilon}{\sigma_1}\right)$$

will satisfy the requirements, and we will get

$$\frac{R}{2^{b-1}-1} = \delta = \mathcal{O}\left(\frac{\varepsilon}{\sigma_1}\right) \quad \Rightarrow \quad b = \log_2 \mathcal{O}\left(\frac{\sigma_1 R}{\varepsilon}\right)$$

This is the expression that we wanted. Notice that even if we did not invoke small $\varepsilon$ in the above big-O analysis, we can set

$$\delta = \mathcal{O}\left(\min\left(\frac{\varepsilon}{\sigma_1}, \frac{\sqrt{\varepsilon\mu}}{L_1}\right)\right)$$

Then our number of bits would look like

$$b = \log_2 \mathcal{O}\left(\max\left(\frac{\sigma_1 R}{\varepsilon}, \frac{RL_1}{\sqrt{\varepsilon\mu}}\right)\right),$$

which shows explicitly that we have replaced the dimension factor with parameters of the loss functions.

## B.3   LP-SGD IN OUR WORK USING NLQ

In Theorem 2, we know that, if $\zeta < \frac{1}{\kappa}$, then

$$\mathbf{E}\left[(f(\tilde{w}) - f(w^*))\right] \leq \frac{1}{2\alpha T}\|w_0 - w^*\|_2^2 + \frac{(1+\eta)\alpha\sigma^2 + \delta\sigma_1 + \zeta\sigma\|w^*\|_2}{2} + \frac{(\delta L_1 + \zeta L\|w^*\|_2 + \zeta\sigma)^2}{4\mu}$$

Set the limit (as $\alpha \to 0$ and $T \to \infty$) to be $\leq \varepsilon$ and replace $\|w^*\|_2$ with $R$; then we get

$$\frac{\delta\sigma_1 + \zeta\sigma R}{2} + \frac{(\delta L_1 + \zeta LR + \zeta\sigma)^2}{4\mu} = \mathcal{O}(\varepsilon).$$

So, in addition to our requirement that $\zeta \leq \kappa^{-1}$, it suffices to have

$$\delta\sigma_1 = \mathcal{O}(\varepsilon), \quad \zeta\sigma R = \mathcal{O}(\varepsilon), \quad \frac{\delta^2 L_1^2}{\mu} = \mathcal{O}(\varepsilon), \quad \frac{\zeta^2(LR+\sigma)^2}{\mu} = \mathcal{O}(\varepsilon).$$

If we set

$$\delta = \frac{\mathcal{O}(\varepsilon)}{\sigma_1}, \quad \zeta = \frac{\mathcal{O}(\varepsilon)}{\sigma R},$$

then all our other requirements will be satisfied for sufficiently small $\varepsilon$. Specifically, we need $\varepsilon$ to be small enough that

$$\frac{\kappa}{\sigma R}\mathcal{O}(\varepsilon) \leq 1, \quad \frac{L_1^2}{\sigma_1^2\mu}\mathcal{O}(\varepsilon) \leq 1, \quad \frac{(LR+\sigma)^2}{\sigma^2 R^2\mu}\mathcal{O}(\varepsilon) \leq 1.$$

As is standard in big-O analysis, we assume that $\varepsilon$ is small enough that these requirements are satisfied, in which case our assignment of $\delta$ and $\zeta$, combined with the results of Theorem 2, is sufficient to ensure an objective gap of $\varepsilon$. Next, starting from (9), the number of bits we need for non-linear quantization must satisfy

$$(1+\zeta)^{(2^{b-1}-1)} - 1 \geq \frac{\zeta R}{\delta}$$

which happens only when

$$\left(2^{b-1} - 1\right)\log(1+\zeta) \geq \log\left(1 + \frac{\zeta R}{\delta}\right).$$

Since we know that $0 \leq \zeta < 1$, it follows that $\log(1+\zeta) \geq \zeta/2$. So in order for the above to be true, it suffices to have

$$\left(2^{b-1} - 1\right) \cdot \frac{\zeta}{2} \geq \log\left(1 + \frac{\zeta R}{\delta}\right).$$

Since $2^{b-1} - 1 > 2^{b-2}$, it follows that it suffices to have

$$2^b \cdot \frac{\zeta}{8} \geq \log\left(1 + \frac{\zeta R}{\delta}\right).$$

And this will be true if

$$b = \log_2 \mathcal{O}\left(\frac{1}{\zeta} \log\left(1 + \frac{\zeta R}{\delta}\right)\right).$$

Finally, using our assignment of $\delta$ and $\zeta$ gives us

$$b = \log_2 \mathcal{O}\left(\frac{\sigma R}{\epsilon} \log\left(1 + \frac{\sigma_1}{\sigma}\right)\right).$$

This is the expression that we wanted. Notice that even if we did not invoke small $\varepsilon$ in the above big-O analysis, we would still get a rate in which all of our $\ell_1$-dependent terms are inside the double-logarithm, because none of the requirements above that constrain $\zeta$ are $\ell_1$-dependent. To be explicit, to do this we would set $\delta$ and $\zeta$ to be

$$\delta = \mathcal{O}\left(\min\left(\frac{\varepsilon}{\sigma_1}, \frac{\sqrt{\varepsilon\mu}}{L_1}\right)\right), \quad \zeta = \mathcal{O}\left(\min\left(\frac{\varepsilon}{\sigma R}, \frac{\sqrt{\varepsilon\mu}}{LR + \sigma}, \frac{1}{\kappa}\right)\right).$$

Then our number of bits would look like

$$b = \log_2 \mathcal{O}\left(\max\left(\frac{\sigma R}{\varepsilon}, \frac{LR + \sigma}{\sqrt{\varepsilon\mu}}, \kappa\right) \cdot \log\left(1 + \frac{\zeta R}{\delta}\right)\right),$$

which shows explicitly that any $\ell_1$-dependent terms are inside the double logarithm.

## C  PROOF FOR THEOREMS

Before we prove the main theorems presented in the paper, we will prove the following lemmas that will be useful later, as well as the lemmas we presented before.

The proof of lemma 1 can be extracted from the proof of lemma 5 that we will show later.

*Proof of Lemma 2.* Here we consider the positive case first, then symmetrically the negative case also holds. First, for normal FPQ, the set of quantization points are:

$$D = \{0\} \cup \left\{ s \cdot \left(1 + \frac{x}{n_m}\right) \cdot 2^y \mid x = 0, 1, \cdots, n_m - 1, \ y = -\frac{n_e}{2} + 2, \cdots, \frac{n_e}{2} - 1 \right\}$$

and we set the parameters for the nonlinear quantization bound to be:

$$\delta = s \cdot 2^{-\frac{n_e}{2}+2} = \frac{4s}{\left(\sqrt{2}\right)^{n_e}}, \quad \zeta = \frac{1}{n_m}, \quad \eta = \frac{\zeta^2}{4(1+\zeta)} = \frac{1}{4n_m(n_m+1)}$$

For any $w$ within representable range, we can assume it is in $[q_i, q_{i+1})$, then

$$\mathbf{E}\left[[Q(w) - w]^2\right] = \frac{q_{i+1} - w}{q_{i+1} - q_i} \cdot (w - q_i)^2 + \frac{w - q_i}{q_{i+1} - q_i} \cdot (q_{i+1} - w)^2$$
$$= (w - q_i)(q_{i+1} - w)$$

So now we only need to prove that

$$\forall v \in D, \ (w - q_i)(q_{i+1} - w) \le \delta \cdot |w - v| + \zeta \cdot |v| \cdot |w - v| + \eta \cdot |w - v|^2$$

First, we consider a special case where $q_i = 0$. In this case, $q_{i+1} = s \cdot 1 \cdot 2^{-\frac{n_e}{2}+2} = \delta$. If $v = 0$, it is obvious that

$$LHS = (w - q_i)(q_{i+1} - w) = w(\delta - w) \le \delta w \le RHS$$

and similarly for $v = \delta$,

$$LHS = (w - q_i)(q_{i+1} - w) = w(\delta - w) \le \delta(\delta - w) \le RHS$$

and for $v > \delta$,

$$RHS \ge \delta(v - w) \ge \delta(\delta - w) \ge w(\delta - w) = LHS$$

Next, we consider the case where $q_i \neq 0$. In this case, we can assume $q_i = s \cdot \left(1 + \frac{x}{n_m}\right) \cdot 2^y$, then $q_{i+1} - q_i = s \cdot 2^y \leq \frac{1}{n_m} q_i = \zeta q_i$.

If $v \geq q_{i+1}$, denote $y = q_{i+1} - w$, then

$$LHS = (w - q_i)(q_{i+1} - w) = y \cdot (q_{i+1} - q_i - y) = y \cdot (\zeta q_i - y)$$
$$RHS \geq \zeta \cdot q_{i+1} \cdot (q_{i+1} - w) = \zeta q_{i+1} y \geq \zeta q_i y \geq LHS$$

Secondly, if $0 \leq v \leq q_i$, denote $y = w - q_i$, then

$$LHS = (w - q_i)(q_{i+1} - w) = y \cdot (q_{i+1} - q_i - y) = y \cdot (\zeta q_i - y)$$
$$RHS = \delta \cdot (w - v) + \zeta \cdot v \cdot (w - v) + \eta \cdot (w - v)^2$$
$$= -(\zeta - \eta) \cdot v^2 + (-\delta + \zeta w - 2\eta w) \cdot v + \delta w - \delta^2$$

observe that $\zeta - \eta > 0$, so the right hand side is a concave function of $v$, thus it achieves minimum at either $v = 0$ or $v = q_i$. At $v = q_i$:

$$RHS = \delta y + \zeta q_i y + \eta y^2 \geq \zeta q_i y \geq LHS$$

and at $v = 0$, since $q_{i+1} \leq (1 + \zeta)q_i$ and $q_i \leq w$,

$$RHS - LHS = \delta \cdot w + \zeta \cdot 0 \cdot w + \eta \cdot w^2 - (w - q_i)(q_{i+1} - w)$$
$$= (1 + \eta)w^2 + (\delta - q_i - q_{i+1})w + q_i q_{i+1}$$
$$\geq (1 + \eta)w^2 - (q_i + q_{i+1}) \cdot w + q_i q_{i+1}$$
$$= (1 + \eta)w^2 - [(2 + \zeta)q_i w + (q_{i+1} - (1 + \zeta)q_i)w] + [(1 + \zeta)q_i^2 + (q_{i+1} - (1 + \zeta)q_i)q_i]$$
$$= (1 + \eta)w^2 - (2 + \zeta)q_i \cdot w + (1 + \zeta)q_i^2 + (q_{i+1} - (1 + \zeta)q_i)(q_i - w)$$
$$\geq (1 + \eta)w^2 - (2 + \zeta)q_i \cdot w + (1 + \zeta)q_i^2$$

which is a positive parabola. Recall that $\eta = \frac{\zeta^2}{4(\zeta+1)} = \frac{(\zeta+2)^2}{4(\zeta+1)} - 1$, thus the determinant is $(2 + \zeta)^2 q_i^2 - 4(1 + \eta)(1 + \zeta)q_i^2 = 0$, therefore $RHS - LHS \geq 0$.

Now we extend this conclusion to the case where $v \leq 0$. In this case,

$$RHS = \delta \cdot (w - v) + \zeta \cdot (-v) \cdot (w - v) + \eta \cdot (w - v)^2$$

since $w, \zeta, \delta, \eta$ are all positive, this is apparently a decreasing function of $v$, thus it achieves minimum at $v = 0$, which is what we have already proven.

So far, we've proven the lemma in the case of $w \geq 0, v \geq 0$ and $w \geq 0, v \leq 0$, and symmetrically it holds for $w \leq 0, v \leq 0$ and $w \leq 0, v \geq 0$, which indicates that we can extend $D$ to be a set containing both positive and negative numbers.

In the de-normal FPQ case, the set of quantization points are:

$$D = \left\{ s \cdot \frac{x}{n_m} \cdot 2^{-\frac{n_e}{2}+3} \mid x = 0, 1, \cdots, n_m - 1 \right\}$$
$$\cup \left\{ s \cdot \left(1 + \frac{x}{n_m}\right) \cdot 2^y \mid x = 0, 1, \cdots, n_m - 1, \ y = -\frac{n_e}{2} + 3, \cdots, \frac{n_e}{2} - 1 \right\}$$

and we set the parameters for the nonlinear quantization bound to be:

$$\delta = s \cdot \frac{1}{n_m} \cdot 2^{-\frac{n_e}{2}+3} = \frac{8}{C} \cdot \frac{s n_e}{\left(\sqrt{2}\right)^{n_e}}, \quad \zeta = \frac{1}{n_m}, \quad \eta = \frac{\zeta^2}{4(1 + \zeta)} = \frac{1}{4 n_m (n_m + 1)}$$

The proof for this case follows the exact same structure as the normal FPQ case. $\qquad \square$

**Lemma 3.** *Under condition of linear quantization when using low-precision representation* $(\delta, b)$, *for any* $w, v \in \mathbb{R}^d$ *where* $Q_{(\delta,b)}(w) = w$,

$$\mathbf{E}\left[\left\|Q_{(\delta,b)}(w + v) - w^*\right\|_2^2\right] \leq \|(w + v) - w^*\|_2^2 + \delta \|v\|_1.$$

*where $Q$ is the linear quantization function.*

*Proof of Lemma 3.* (This proof follows the same structure as the proof for lemma 1 in (De Sa et al., 2018)) First, observe that this lemma holds if it holds for each dimension, so we only need to prove that for any $w, v \in \mathbb{R}$ where $Q_{(\delta,b)}(w) = w$, i.e. $w \in \text{dom}(\delta, b)$,

$$\mathbf{E}\left[(Q_{(\delta,b)}(w+v) - w^*)^2\right] \leq (w+v-w^*)^2 + \delta|v|$$

then we can sum up all the dimensions to get the result.

Now we consider the problem in two situations. First, if $w + v$ is within the range representable by $(\delta, b)$, then $\mathbf{E}\left[Q_{(\delta,b)}(w+v)\right] = w + v$. In this case,

$$
\begin{aligned}
&\mathbf{E}\left[(Q_{(\delta,b)}(w+v) - w^*)^2\right] \\
&= \mathbf{E}\left[[(Q_{(\delta,b)}(w+v) - (w+v)) - ((w+v) - w^*)]^2\right] \\
&= \mathbf{E}\left[[Q_{(\delta,b)}(w+v) - (w+v)]^2 - 2[Q_{(\delta,b)}(w+v) - (w+v)][(w+v) - w^*]\right] \\
&\quad + [(w+v) - w^*]^2 \\
&= \mathbf{E}\left[[Q_{(\delta,b)}(w+v) - (w+v)]^2\right] - 2[(w+v) - (w+v)][(w+v) - w^*] \\
&\quad + [(w+v) - w^*]^2 \\
&= [(w+v) - w^*]^2 + \mathbf{E}\left[[Q_{(\delta,b)}(w+v) - (w+v)]^2\right]
\end{aligned}
$$

Since $(w + v)$ is within representable range, $\mathbf{E}\left[[Q_{(\delta,b)}(w+v) - (w+v)]^2\right]$ is equivalent to $\mathbf{E}\left[[Q_{(\delta,\infty)}(v) + w - (w+v)]^2\right]$, which equals $\mathbf{E}\left[[Q_{(\delta,\infty)}(v) - v]^2\right]$ since $Q_{(\delta,b)}(w) = w$.

Now we only need to prove that $\mathbf{E}\left[[Q_{(\delta,\infty)}(v) - v]^2\right] \leq \delta|v|$. Observe that this trivially holds for $v = 0$, and is symmetrical for positive and negative $v$. Without loss of generality we assume $v > 0$, let $z$ be the rounded-down quantization of $v$, then we have $z \geq 0$. Then $Q_{(\delta,b)}(v)$ will round to $z+\delta$ (the rounded-up quantization of $v$) with probability $\frac{v-z}{\delta}$, and it will round to $z$ with probability $\frac{z+\delta-v}{\delta}$. This quantization is unbiased because

$$\mathbf{E}\left[Q_{(\delta,\infty)}(w)\right] = \frac{v-z}{\delta}(z+\delta) + \frac{z+\delta-v}{\delta}z = \frac{vz - z^2 + v\delta - z\delta}{\delta} + \frac{z^2 + z\delta - vz}{\delta} = v.$$

Thus, its variance will be

$$
\begin{aligned}
\mathbf{E}\left[(Q_{(\delta,\infty)}(v) - v)^2\right] &= \frac{v-z}{\delta}(z+\delta-v)^2 + \frac{z+\delta-v}{\delta}(z-v)^2 \\
&= (v-z)(z+\delta-v)\left(\frac{z+\delta-v}{\delta} + \frac{v-z}{\delta}\right) \\
&= \delta(v-z) - (v-z)^2 \\
&\leq \delta(v-z) \leq \delta v.
\end{aligned}
$$

therefore

$$\mathbf{E}\left[(Q_{(\delta,b)}(w+v) - w^*)^2\right] \leq (w+v-w^*)^2 + \delta|v|$$

In the other case, when $w + v$ is on the exterior of the representable region, the quantization function $Q_{(\delta,b)}$ just maps it to the nearest representable value. Since $w^*$ is in the interior of the representable region, this operation will make $w + v$ closer to $w^*$. Thus,

$$(Q_{(\delta,b)}(w+v) - w^*)^2 \leq (w+v-w^*)^2,$$

and so it will certainly be the case that

$$\mathbf{E}\left[(Q_{(\delta,b)}(w+v) - w^*)^2\right] \leq (w+v-w^*)^2 + \delta|v|.$$

Now that we've proven the inequality for one dimension, we can sum up all $d$ dimensions and get

$$\mathbf{E}\left[\left\|Q_{(\delta,b)}(w+v) - w^*\right\|_2^2\right] \leq \|(w+v) - w^*\|_2^2 + \delta\|v\|_1.$$

$\square$

For completeness, we also re-state the proof of following lemma, which was presented as equation (8) in (Johnson and Zhang, 2013), and here we present the proof for this lemma used in (De Sa et al., 2018).

**Lemma 4.** *Under the standard condition of Lipschitz continuity, if $i$ is sampled uniformly at random from $\{1, \dots, N\}$, then for any $w$,*

$$\mathbf{E}\left[\|\nabla f_i(w) - \nabla f_i(w^*)\|_2^2\right] \leq 2L\left(f(w) - f(w^*)\right).$$

*Proof of Lemma 4.* For any $i$, define

$$g_i(w) = f_i(w) - f_i(w^*) - (w - w^*)^T \nabla f_i(w^*).$$

Clearly, if $i$ is sampled randomly as in the lemma statement, $\mathbf{E}\left[g_i(w)\right] = f(w)$. But also, $w^*$ must be the minimizer of $g_i$, so for any $w$

$$
\begin{aligned}
g_i(w^*) &\leq \min_\eta g_i(w - \eta \nabla g_i(w)) \\
&\leq \min_\eta \left(g_i(w) - \eta \|\nabla g_i(w)\|_2^2 + \frac{\eta^2 L}{2} \|\nabla g_i(w)\|_2^2\right) \\
&= g_i(w) - \frac{1}{2L} \|\nabla g_i(w)\|_2^2.
\end{aligned}
$$

where the second inequality follows from the Lipschitz continuity property. Re-writing this in terms of $f_i$ and averaging over all the $i$ now proves the lemma statement. $\qquad\square$

**Lemma 5.** *Under the condition of logarithmic quantization, for any $w, v \in \mathbb{R}^d$ where $v \in D^d$,*

$$\mathbf{E}\left[\|Q(w) - w^*\|_2^2\right] \leq \|w - w^*\|_2^2 + \delta \|w - v\|_1 + \zeta \|v\|_2 \|w - v\|_2 + \eta \|w - v\|_2^2$$

*where $Q$ is the non-linear quantization function.*

Note that the proof this lemma naturally extends to lemma 1, thus we omitted the proof for lemma 1 and just present the proof for lemma 5.

*Proof of Lemma 5.* Here we only consider the positive case first, where

$$D = \{q_0, q_1, \cdots, q_{n-1}\}$$

with $[0, q_{n-1}]$ being the representable range of $D$. As for the negative case, we will show later that it holds symmetrically.

Observe that this lemma holds if it holds for each dimension, so we only need to prove that for any $w, v \in \mathbb{R}$ where $v \in D$,

$$\mathbf{E}\left[[Q(w) - w^*]^2\right] \leq |w - w^*|^2 + \delta \cdot |w - v| + \zeta \cdot |v| \cdot |w - v| + \eta \cdot |w - v|^2$$

then we can sum up all the dimensions and use Cauchy-Schwarz inequality to get the result.

Now we consider the problem in two situations.

First, if $w$ is outside the representable range, the quantization function $Q$ just maps it to the nearest representable value. Since $w^*$ is in the interior of the representable range, this operation will make $w$ closer to $w^*$. Thus,

$$[Q(w) - w^*]^2 \leq (w - w^*)^2,$$

and so it will certainly be the case that

$$\mathbf{E}\left[[Q(w) - w^*]^2\right] \leq |w - w^*|^2 + \delta \cdot |w - v| + \zeta \cdot |v| \cdot |w - v| + \eta \cdot |w - v|^2$$

Second, if $w$ is within the representable range, then $\mathbf{E}\left[Q(w)\right] = w$. In this case,

$$
\begin{aligned}
&\mathbf{E}\left[[Q(w) - w^*]^2\right] \\
={}& \mathbf{E}\left[[(Q(w) - w) - (w - w^*)]^2\right] \\
={}& \mathbf{E}\left[[Q(w) - w]^2 - 2[Q(w) - w](w - w^*)\right] + (w - w^*)^2 \\
={}& \mathbf{E}\left[[Q(w) - w]^2\right] - 2(w - w)(w - w^*) + (w - w^*)^2 \\
={}& (w - w^*)^2 + \mathbf{E}\left[[Q(w) - w]^2\right]
\end{aligned}
$$

Since $w$ is within representable range, we can assume it is in $[q_i, q_{i+1})$, then

$$\mathbf{E}\left[[Q(w) - w]^2\right] = \frac{q_{i+1} - w}{q_{i+1} - q_i} \cdot (w - q_i)^2 + \frac{w - q_i}{q_{i+1} - q_i} \cdot (q_{i+1} - w)^2$$
$$= (w - q_i)(q_{i+1} - w)$$

So now we only need to prove that

$$(w - q_i)(q_{i+1} - w) \leq \delta \cdot |w - v| + \zeta \cdot |v| \cdot |w - v| + \eta \cdot |w - v|^2$$

Note that $v \in D$, so it is either $v \geq q_{i+1}$ or $v \leq q_i$.

Firstly, if $v \geq q_{i+1}$, denote $y = q_{i+1} - w$, then

$$\begin{aligned} LHS = & (w - q_i)(q_{i+1} - w) = y \cdot (q_{i+1} - q_i - y) = y \cdot (\delta + \zeta q_i - y) \\ RHS = & \delta \cdot (v - w) + \zeta \cdot v \cdot (v - w) + \eta \cdot (v - w)^2 \\ \geq & \delta \cdot (q_{i+1} - w) + \zeta \cdot q_{i+1} \cdot (q_{i+1} - w) + \eta \cdot (q_{i+1} - w)^2 \\ = & \delta y + \zeta q_{i+1} y + \eta y^2 \\ \geq & \delta y + \zeta q_i y - y^2 = LHS \end{aligned}$$

Secondly, if $0 \leq v \leq q_i$, denote $y = w - q_i$, then

$$\begin{aligned} LHS = & (w - q_i)(q_{i+1} - w) = y \cdot (q_{i+1} - q_i - y) = y \cdot (\delta + \zeta q_i - y) \\ RHS = & \delta \cdot (w - v) + \zeta \cdot v \cdot (w - v) + \eta \cdot (w - v)^2 \\ = & -(\zeta - \eta) \cdot v^2 + (-\delta + \zeta w - 2\eta w) \cdot v + \delta w - \delta^2 \end{aligned}$$

observe that $\zeta - \eta > 0$, so the right hand side is a concave function of $v$, thus it achieves minimum at either $v = 0$ or $v = q_i$. At $v = q_i$:

$$RHS = \delta y + \zeta q_i y + \eta y^2 \geq \delta y + \zeta q_i y - y^2 = LHS$$

and at $v = 0$:

$$\begin{aligned} RHS - LHS = & \delta \cdot w + \zeta \cdot 0 \cdot w + \eta \cdot w^2 - (w - q_i)(q_{i+1} - w) \\ = & (1 + \eta)w^2 + (\delta - q_i - q_{i+1})w + q_i q_{i+1} \\ = & (1 + \eta)w^2 - (2 + \zeta)q_i \cdot w + q_i q_{i+1} \\ \geq & (1 + \eta)w^2 - (2 + \zeta)q_i \cdot w + (1 + \zeta)q_i^2 \end{aligned}$$

which is a positive parabola. Recall that $\eta = \frac{\zeta^2}{4(\zeta+1)} = \frac{(\zeta+2)^2}{4(\zeta+1)} - 1$, thus the determinant is $(2 + \zeta)^2 q_i^2 - 4(1 + \eta)(1 + \zeta)q_i^2 = 0$, therefore $RHS - LHS \geq 0$.

Now we extend this conclusion to the case where $v \leq 0$. In this case,

$$RHS = \delta \cdot (w - v) + \zeta \cdot (-v) \cdot (w - v) + \eta \cdot (w - v)^2$$

since $w, \zeta, \delta, \eta$ are all positive, this is apparently a decreasing function of $v$, thus it achieves minimum at $v = 0$, which is what we have already proven.

So far, we've proven the lemma in the case of $w \geq 0, v \geq 0$ and $w \geq 0, v \leq 0$, and symmetrically it holds for $w \leq 0, v \leq 0$ and $w \leq 0, v \geq 0$, which indicates that we can extend $D$ to be a set containing both positive and negative numbers, and we can reset $D$ to be

$$D = \{-q_n, \cdots, -q_1, q_0, q_1, \cdots, q_{n-1}\}$$

where

$$q_0 = 0, \quad q_{i+1} - q_i = \delta + \zeta q_i$$

$\square$

Now we have proven all the lemmas we need. Next, we make some small modifications to the assumptions (weakening them) so that our theorems are shown in a more general sense. For assumption 2, we change it to:

**Assumption 5.** *All the gradients of the loss functions $f_i$ are $L_1$-Lipschitz continuous in the sense of 1-norm to p-norm, that is,*

$$\forall i \in \{1, 2, \cdots n\}, \quad \forall x, y, \quad \|\nabla f_i(x) - \nabla f_i(y)\|_1 \leq L_1 \|x - y\|_p$$

While in the body of the paper and in our experiments we choose $p = 2$ for simplicity, here we are going to prove that a generalization of Theorem 1 holds for all real numbers $p$. We also need a similar generalization of Assumption 3.

**Assumption 6.** *The average of the loss functions $f = \frac{1}{n} \sum_i f_i$ is $\mu_1-$ strongly convex near the optimal point in the sense of p-norm, that is,*

$$\forall w, \quad \frac{\mu_1}{2} \|w - w^*\|_p^2 \leq f(w) - f(w^*)$$

*with p being any real number.*

This assumption is essentially the same as the assumption for strong convexity that we stated before, since in practice we would choose $p = 2$ and then $\mu_1$ and $\mu$ would be the same. But here we are actually presenting our result in a stronger sense in that we can choose any real number $p$ and the proof goes the same.

Now we are ready to prove the theorems. Note that the result of the following proof contains $\mu_1$ since we are proving a more general version of our theorems; substituting them with $\mu$ will lead to the same result that we stated before.

*Proof of Theorem 1.* In low-precision SGD, we have:

$$u_{t+1} = w_t - \alpha \nabla \tilde{f}_t(w_t), \quad w_{t+1} = Q(u_{t+1})$$

by lemma 3, we know that

$$
\begin{aligned}
\mathbf{E}\left[\|w_{t+1} - w^*\|_2^2\right] &= \mathbf{E}\left[\left\|Q(w_t - \alpha \nabla \tilde{f}_t(w_t)) - w^*\right\|_2^2\right] \\
&\leq \mathbf{E}\left[\left\|w_t - \alpha \nabla \tilde{f}_t(w_t) - w^*\right\|_2^2\right] + \delta \mathbf{E}\left[\left\|\alpha \nabla \tilde{f}_t(w_t)\right\|_1\right] \\
&= \mathbf{E}\left[\|w_t - w^*\|_2^2\right] - 2\alpha \mathbf{E}\left[(w_t - w^*)^T \nabla \tilde{f}_t(w_t)\right] \\
&\quad + \alpha^2 \mathbf{E}\left[\left\|\nabla \tilde{f}_t(w_t)\right\|_2^2\right] + \alpha \delta \mathbf{E}\left[\left\|\nabla \tilde{f}_t(w_t)\right\|_1\right] \\
&\leq \mathbf{E}\left[\|w_t - w^*\|_2^2\right] - 2\alpha \mathbf{E}\left[(f(w_t) - f(w^*)) + \frac{\mu}{2}\|w_t - w^*\|_2^2\right] \\
&\quad + \alpha^2 \mathbf{E}\left[\left\|\nabla \tilde{f}_t(w_t)\right\|_2^2\right] + \alpha \delta \mathbf{E}\left[\left\|\nabla \tilde{f}_t(w_t)\right\|_1\right] \\
&= (1 - \alpha\mu)\mathbf{E}\left[\|w_t - w^*\|_2^2\right] + \alpha^2 \mathbf{E}\left[\left\|\nabla \tilde{f}_t(w_t)\right\|_2^2\right] + \alpha \delta \mathbf{E}\left[\left\|\nabla \tilde{f}_t(w_t)\right\|_1\right] \\
&\quad - 2\alpha \mathbf{E}\left[(f(w_t) - f(w^*))\right]
\end{aligned}
$$

where the second inequality holds due to the strongly convexity assumption. According to the assumptions we had, we have:

$$
\begin{aligned}
\mathbf{E}\left[\|\nabla f_i(w)\|_2^2\right] &= \mathbf{E}\left[\|\nabla f_i(w) - \nabla f_i(w^*) + \nabla f_i(w^*)\|_2^2\right] \\
&= \mathbf{E}\left[\|\nabla f_i(w) - \nabla f_i(w^*)\|_2^2 + 2(\nabla f_i(w) - \nabla f_i(w^*))^T \nabla f_i(w^*) + \|\nabla f_i(w^*)\|_2^2\right] \\
&= \mathbf{E}\left[\|\nabla f_i(w) - \nabla f_i(w^*)\|_2^2 + \|\nabla f_i(w^*)\|_2^2\right] \\
&\leq L^2 \cdot \mathbf{E}\left[\|w - w^*\|_2^2\right] + \sigma^2 \\
\mathbf{E}\left[\|\nabla f_i(w)\|_1\right] &= \mathbf{E}\left[\|\nabla f_i(w) - \nabla f_i(w^*) + \nabla f_i(w^*)\|_1\right] \\
&\leq \mathbf{E}\left[\|\nabla f_i(w) - \nabla f_i(w^*)\|_1 + \|\nabla f_i(w^*)\|_1\right] \\
&\leq L_1 \cdot \mathbf{E}\left[\|w - w^*\|_2\right] + \sigma_1
\end{aligned}
$$

where the last inequality holds due to assumption 2 where we let $p = 2$. Applying this result to the previous formula and we will have:

$$
\begin{aligned}
\mathbf{E}\left[\|w_{t+1} - w^*\|_2^2\right] \leq & (1 - \alpha\mu)\mathbf{E}\left[\|w_t - w^*\|_2^2\right] + \alpha^2\mathbf{E}\left[\left\|\nabla\tilde{f}_t(w_t)\right\|_2^2\right] + \alpha\delta\mathbf{E}\left[\left\|\nabla\tilde{f}_t(w_t)\right\|_1\right] \\
& - 2\alpha\mathbf{E}\left[(f(w_t) - f(w^*))\right] \\
\leq & (1 - \alpha\mu + \alpha^2 L^2)\mathbf{E}\left[\|w_t - w^*\|_2^2\right] + \alpha\delta L_1\mathbf{E}\left[\|w_t - w^*\|_2\right] \\
& - 2\alpha\mathbf{E}\left[(f(w_t) - f(w^*))\right] + \alpha^2\sigma^2 + \alpha\delta\sigma_1
\end{aligned}
$$

Here we introduce a positive constant $C$ that we'll set later, and by basic inequality we get

$$
\alpha\delta L_1\mathbf{E}\left[\|w_t - w^*\|_2\right] \leq C\mathbf{E}\left[\|w_t - w^*\|_2\right]^2 + \frac{\alpha^2\delta^2 L_1^2}{4C} \leq C\mathbf{E}\left[\|w_t - w^*\|_2^2\right] + \frac{\alpha^2\delta^2 L_1^2}{4C}
$$

thus

$$
\begin{aligned}
\mathbf{E}\left[\|w_{t+1} - w^*\|_2^2\right] \leq & (1 - \alpha\mu + \alpha^2 L^2 + C)\mathbf{E}\left[\|w_t - w^*\|_2^2\right] - 2\alpha\mathbf{E}\left[(f(w_t) - f(w^*))\right] \\
& + \alpha^2\sigma^2 + \alpha\delta\sigma_1 + \frac{\alpha^2\delta^2 L_1^2}{4C}
\end{aligned}
$$

one setting $C$ to be $\alpha\mu - \alpha^2 L^2$, we will have:

$$
2\alpha\mathbf{E}\left[(f(w_t) - f(w^*))\right] \leq \mathbf{E}\left[\|w_t - w^*\|_2^2\right] - \mathbf{E}\left[\|w_{t+1} - w^*\|_2^2\right] + \alpha^2\sigma^2 + \alpha\delta\sigma_1 + \frac{\alpha^2\delta^2 L_1^2}{4(\alpha\mu - \alpha^2 L^2)}
$$

since we can set $\alpha$ to be small enough such that $\alpha L^2 \leq \frac{\mu}{2}$, then the result will become:

$$
2\alpha\mathbf{E}\left[(f(w_t) - f(w^*))\right] \leq \mathbf{E}\left[\|w_t - w^*\|_2^2\right] - \mathbf{E}\left[\|w_{t+1} - w^*\|_2^2\right] + \alpha^2\sigma^2 + \alpha\delta\sigma_1 + \frac{\alpha\delta^2 L_1^2}{2\mu}
$$

now we sum up this inequality from $t = 0$ to $t = T - 1$ and divide by $2\alpha T$, then we get:

$$
\begin{aligned}
\frac{1}{T}\sum_{t=o}^{T-1}\mathbf{E}\left[(f(w_t) - f(w^*))\right] \leq & \frac{\|w_0 - w^*\|_2^2 - \mathbf{E}\left[\|w_T - w^*\|_2^2\right]}{2\alpha T} + \frac{\alpha\sigma^2 + \delta\sigma_1}{2} + \frac{\delta^2 L_1^2}{4\mu} \\
\leq & \frac{\|w_0 - w^*\|_2^2}{2\alpha T} + \frac{\alpha\sigma^2 + \delta\sigma_1}{2} + \frac{\delta^2\kappa_1^2\mu}{4}
\end{aligned}
$$

and since we sample $\tilde{w}$ uniformly from $(w_o, w_1, \cdots, w_{T-1})$, we get

$$
\mathbf{E}\left[(f(\tilde{w}) - f(w^*))\right] \leq \frac{1}{2\alpha T}\|w_0 - w^*\|_2^2 + \frac{\alpha\sigma^2 + \delta\sigma_1}{2} + \frac{\delta^2\kappa_1^2\mu}{4}
$$

$\square$

*Proof of Theorem 2.* In low-precision SGD, we have:

$$
u_{t+1} = w_t - \alpha\nabla\tilde{f}_t(w_t), \quad w_{t+1} = Q(u_{t+1})
$$

by lemma 5, we know that

$$
\mathbf{E}\left[\|w_{t+1} - w^*\|_2^2\right] = \mathbf{E}\left[\left\|Q(w_t - \alpha\nabla\tilde{f}_t(w_t)) - w^*\right\|_2^2\right]
$$

$$
\leq \mathbf{E}\left[\left\|w_t - \alpha\nabla\tilde{f}_t(w_t) - w^*\right\|_2^2\right]
$$

$$
+ \delta\mathbf{E}\left[\left\|\alpha\nabla\tilde{f}_t(w_t)\right\|_1\right] + \zeta\mathbf{E}\left[\|w_t\|_2\left\|\alpha\nabla\tilde{f}_t(w_t)\right\|_2\right] + \eta\mathbf{E}\left[\left\|\alpha\nabla\tilde{f}_t(w_t)\right\|_2^2\right]
$$

$$
\leq \mathbf{E}\left[\|w_t - w^*\|_2^2\right] - 2\alpha\mathbf{E}\left[(w_t - w^*)^T\nabla\tilde{f}_t(w_t)\right] + \alpha^2\mathbf{E}\left[\left\|\nabla\tilde{f}_t(w_t)\right\|_2^2\right]
$$

$$
+ \alpha\delta\mathbf{E}\left[\left\|\nabla\tilde{f}_t(w_t)\right\|_1\right] + \zeta\mathbf{E}\left[(\|w_t - w^*\|_2 + \|w^*\|_2)\cdot\alpha\left\|\nabla\tilde{f}_t(w_t)\right\|_2\right] + \eta\alpha^2\mathbf{E}\left[\left\|\nabla\tilde{f}_t(w_t)\right\|_2^2\right]
$$

$$
\leq \mathbf{E}\left[\|w_t - w^*\|_2^2\right] - 2\alpha\mathbf{E}\left[(f(w_t) - f(w^*)) + \frac{\mu}{2}\|w_t - w^*\|_2^2\right] + \alpha^2\mathbf{E}\left[\left\|\nabla\tilde{f}_t(w_t)\right\|_2^2\right]
$$

$$
+ \alpha\delta\mathbf{E}\left[\left\|\nabla\tilde{f}_t(w_t)\right\|_1\right] + \alpha\zeta\mathbf{E}\left[(\|w_t - w^*\|_2 + \|w^*\|_2)\left\|\nabla\tilde{f}_t(w_t)\right\|_2\right] + \eta\alpha^2\mathbf{E}\left[\left\|\nabla\tilde{f}_t(w_t)\right\|_2^2\right]
$$

$$
= (1 - \alpha\mu)\mathbf{E}\left[\|w_t - w^*\|_2^2\right] + (1 + \eta)\alpha^2\mathbf{E}\left[\left\|\nabla\tilde{f}_t(w_t)\right\|_2^2\right] - 2\alpha\mathbf{E}\left[(f(w_t) - f(w^*))\right]
$$

$$
+ \alpha\zeta\mathbf{E}\left[(\|w_t - w^*\|_2 + \|w^*\|_2)\left\|\nabla\tilde{f}_t(w_t)\right\|_2\right] + \alpha\delta\mathbf{E}\left[\left\|\nabla\tilde{f}_t(w_t)\right\|_1\right]
$$

where the third inequality holds due to the strongly convexity assumption. According to the assumptions we had, we have:

$$
\mathbf{E}\left[\|\nabla f_i(w)\|_2^2\right] = \mathbf{E}\left[\|\nabla f_i(w) - \nabla f_i(w^*) + \nabla f_i(w^*)\|_2^2\right]
$$

$$
= \mathbf{E}\left[\|\nabla f_i(w) - \nabla f_i(w^*)\|_2^2 + 2(\nabla f_i(w) - \nabla f_i(w^*))^T\nabla f_i(w^*) + \|\nabla f_i(w^*)\|_2^2\right]
$$

$$
= \mathbf{E}\left[\|\nabla f_i(w) - \nabla f_i(w^*)\|_2^2 + \|\nabla f_i(w^*)\|_2^2\right]
$$

$$
\leq L^2\cdot\mathbf{E}\left[\|w - w^*\|_2^2\right] + \sigma^2
$$

$$
\|\nabla f_i(w)\|_2 = \|\nabla f_i(w) - \nabla f_i(w^*) + \nabla f_i(w^*)\|_2
$$

$$
\leq \|\nabla f_i(w) - \nabla f_i(w^*)\|_2 + \|\nabla f_i(w^*)\|_2
$$

$$
\leq L\cdot\|w - w^*\|_2 + \sigma
$$

$$
\|\nabla f_i(w)\|_1 = \|\nabla f_i(w) - \nabla f_i(w^*) + \nabla f_i(w^*)\|_1
$$

$$
\leq \|\nabla f_i(w) - \nabla f_i(w^*)\|_1 + \|\nabla f_i(w^*)\|_1
$$

$$
\leq L_1\cdot\|w - w^*\|_2 + \sigma_1
$$

where the last inequality holds due to assumption 2 where we let $p = 2$. Apply this result to the previous formula, denote $\eta' = 1 + \eta$, and then we will have:

$$
\mathbf{E}\left[\|w_{t+1} - w^*\|_2^2\right]
$$

$$
\leq (1 - \alpha\mu)\mathbf{E}\left[\|w_t - w^*\|_2^2\right] + \eta'\alpha^2\mathbf{E}\left[\left\|\nabla\tilde{f}_t(w_t)\right\|_2^2\right] - 2\alpha\mathbf{E}\left[(f(w_t) - f(w^*))\right]
$$

$$
+ \alpha\zeta\mathbf{E}\left[(\|w_t - w^*\|_2 + \|w^*\|_2)\left\|\nabla\tilde{f}_t(w_t)\right\|_2\right] + \alpha\delta\mathbf{E}\left[\left\|\nabla\tilde{f}_t(w_t)\right\|_1\right]
$$

$$
\leq (1 - \alpha\mu + \eta'\alpha^2 L^2)\mathbf{E}\left[\|w_t - w^*\|_2^2\right] + \alpha\delta L_1\mathbf{E}\left[\|w_t - w^*\|_2\right] + \eta'\alpha^2\sigma^2 + \alpha\delta\sigma_1
$$

$$
+ \alpha\zeta\mathbf{E}\left[(\|w_t - w^*\|_2 + \|w^*\|_2)(L\cdot\|w - w^*\|_2 + \sigma)\right] - 2\alpha\mathbf{E}\left[(f(w_t) - f(w^*))\right]
$$

$$
= (1 - \alpha\mu + \alpha\zeta L + \eta'\alpha^2 L^2)\mathbf{E}\left[\|w_t - w^*\|_2^2\right] - 2\alpha\mathbf{E}\left[(f(w_t) - f(w^*))\right]
$$

$$
+ (\alpha\delta L_1 + \alpha\zeta L\|w^*\|_2 + \alpha\zeta\sigma)\mathbf{E}\left[\|w_t - w^*\|_2\right] + \eta'\alpha^2\sigma^2 + \alpha\delta\sigma_1 + \alpha\zeta\sigma\|w^*\|_2
$$

Here we introduce a positive constant $C$ that we'll set later, and by basic inequality we get

$$(\alpha\delta L_1 + \alpha\zeta L \|w^*\|_2 + \alpha\zeta\sigma)\mathbf{E}\left[\|w_t - w^*\|_2\right]$$

$$\leq C\mathbf{E}\left[\|w_t - w^*\|_2\right]^2 + \frac{(\alpha\delta L_1 + \alpha\zeta L \|w^*\|_2 + \alpha\zeta\sigma)^2}{4C}$$

$$\leq C\mathbf{E}\left[\|w_t - w^*\|_2^2\right] + \frac{\alpha^2(\delta L_1 + \zeta L \|w^*\|_2 + \zeta\sigma)^2}{4C}$$

thus

$$\mathbf{E}\left[\|w_{t+1} - w^*\|_2^2\right] \leq (1 - \alpha\mu + \alpha\zeta L + \eta'\alpha^2 L^2 + C)\mathbf{E}\left[\|w_t - w^*\|_2^2\right] - 2\alpha\mathbf{E}\left[(f(w_t) - f(w^*))\right]$$

$$+ \eta'\alpha^2\sigma^2 + \alpha\delta\sigma_1 + \alpha\zeta\sigma \|w^*\|_2 + \frac{\alpha^2(\delta L_1 + \zeta L \|w^*\|_2 + \zeta\sigma)^2}{4C}$$

one setting $C$ to be $\alpha\mu - \alpha\zeta L - \eta'\alpha^2 L^2$, we will have:

$$2\alpha\mathbf{E}\left[(f(w_t) - f(w^*))\right] \leq \mathbf{E}\left[\|w_t - w^*\|_2^2\right] - \mathbf{E}\left[\|w_{t+1} - w^*\|_2^2\right]$$

$$+ \eta'\alpha^2\sigma^2 + \alpha\delta\sigma_1 + \alpha\zeta\sigma \|w^*\|_2 + \frac{\alpha^2(\delta L_1 + \zeta L \|w^*\|_2 + \zeta\sigma)^2}{4(\alpha\mu - \alpha\zeta L - \eta'\alpha^2 L^2)}$$

since we can set $\alpha$ to be small enough such that $\alpha\mu - \alpha\zeta L - \eta'\alpha^2 L^2 \geq \frac{1}{2}\alpha\mu$, then the result will become:

$$2\alpha\mathbf{E}\left[(f(w_t) - f(w^*))\right] \leq \mathbf{E}\left[\|w_t - w^*\|_2^2\right] - \mathbf{E}\left[\|w_{t+1} - w^*\|_2^2\right]$$

$$+ \eta'\alpha^2\sigma^2 + \alpha\delta\sigma_1 + \alpha\zeta\sigma \|w^*\|_2 + \frac{\alpha(\delta L_1 + \zeta L \|w^*\|_2 + \zeta\sigma)^2}{2\mu}$$

now we sum up this inequality from $t = 0$ to $t = T - 1$ and divide by $2\alpha T$, then we get:

$$\frac{1}{T}\sum_{t=o}^{T-1} \mathbf{E}\left[(f(w_t) - f(w^*))\right]$$

$$\leq \frac{\|w_0 - w^*\|_2^2 - \mathbf{E}\left[\|w_T - w^*\|_2^2\right]}{2\alpha T} + \frac{\eta'\alpha\sigma^2 + \delta\sigma_1 + \zeta\sigma \|w^*\|_2}{2} + \frac{(\delta L_1 + \zeta L \|w^*\|_2 + \zeta\sigma)^2}{4\mu}$$

$$\leq \frac{\|w_0 - w^*\|_2^2}{2\alpha T} + \frac{\eta'\alpha\sigma^2 + \delta\sigma_1 + \zeta\sigma \|w^*\|_2}{2} + \frac{(\delta L_1 + \zeta L \|w^*\|_2 + \zeta\sigma)^2}{4\mu}$$

and since we sample $\tilde{w}$ uniformly from $(w_o, w_1, \cdots, w_{T-1})$, we get

$$\mathbf{E}\left[(f(\tilde{w}) - f(w^*))\right] \leq \frac{1}{2\alpha T}\|w_0 - w^*\|_2^2 + \frac{\eta'\alpha\sigma^2 + \delta\sigma_1 + \zeta\sigma \|w^*\|_2}{2} + \frac{(\delta L_1 + \zeta L \|w^*\|_2 + \zeta\sigma)^2}{4\mu}$$

$$\square$$

*Proof of Theorem 3.* In the normal FPQ case, the set of quantization points are:

$$D = \{0\} \cup \left\{s \cdot \left(1 + \frac{x}{n_m}\right) \cdot 2^y \mid x = 0, 1, \cdots, n_m - 1, \ y = -\frac{n_e}{2} + 2, \cdots, \frac{n_e}{2} - 1\right\}$$

then the parameters for the nonlinear quantization bound can be computed as:

$$\delta = s \cdot 2^{-\frac{n_e}{2}+2} = \frac{4s}{(\sqrt{2})^{n_e}}, \quad \zeta = \frac{1}{n_m}, \quad \eta = \frac{\zeta^2}{4(1+\zeta)} = \frac{1}{4n_m(n_m + 1)}$$

For NLQ-SGD, the noise ball size according to theorem 2 is:

$$\frac{\delta\sigma_1 + \zeta\sigma \|w^*\|_2}{2} + \frac{(\delta L_1 + \zeta L \|w^*\|_2 + \zeta\sigma)^2}{4\mu}$$

Denote this as $\frac{1}{2}A + \frac{1}{4\mu}B^2$. When $b$ is large, $\delta, \zeta, \eta$ are small, then the dominating term for the noise ball is

$$A = \delta\sigma_1 + \zeta\sigma \|w^*\|_2 = 4s\sigma_1 \frac{1}{(\sqrt{2})^{n_e}} + \sigma \|w^*\|_2 \frac{1}{n_m} = 4s\sigma_1 \frac{1}{(\sqrt{2})^{n_e}} + \sigma \|w^*\|_2 \frac{n_e}{C}$$

let the derivative over $n_e$ to be 0 and we get:

$$\frac{\partial A}{\partial n_e} = -2(\ln 2)s\sigma_1 \frac{1}{(\sqrt{2})^{n_e}} + \sigma \|w^*\|_2 \frac{1}{C} = 0, \quad \left(\sqrt{2}\right)^{n_e} = \frac{2(\ln 2)s\sigma_1 C}{\sigma \|w^*\|_2}$$

$$n_e = 2\log_2\left(\frac{2(\ln 2)s\sigma_1 C}{\sigma \|w^*\|_2}\right), \quad b_e = \log_2\left(2b + 2\log_2\left(\frac{2(\ln 2)s\sigma_1}{\sigma \|w^*\|_2}\right)\right)$$

And when $b$ is small, $\delta, \zeta, \eta$ are large and the dominating term for the noise ball is

$$B = \delta L_1 + \zeta L \|w^*\|_2 + \zeta\sigma = 4sL_1 \frac{1}{(\sqrt{2})^{n_e}} + (L \|w^*\|_2 + \sigma)\frac{1}{n_m} = 4sL_1 \frac{1}{(\sqrt{2})^{n_e}} + (L \|w^*\|_2 + \sigma)\frac{n_e}{C}$$

let the derivative of $n_e$ to be 0 and we get:

$$\frac{\partial B}{\partial n_e} = -2(\ln 2)sL_1 \frac{1}{(\sqrt{2})^{n_e}} + (L \|w^*\|_2 + \sigma)\frac{1}{C} = 0, \quad \left(\sqrt{2}\right)^{n_e} = \frac{2(\ln 2)sL_1 C}{L \|w^*\|_2 + \sigma}$$

$$n_e = 2\log_2\left(\frac{2(\ln 2)sL_1 C}{L \|w^*\|_2 + \sigma}\right), \quad b_e = \log_2\left(2b + 2\log_2\left(\frac{2(\ln 2)sL_1}{L \|w^*\|_2 + \sigma}\right)\right)$$

For $b$ such that neither the terms dominates the result, we know the noise ball size is:

$$\frac{1}{2}A + \frac{1}{4\mu}B^2 = \frac{\delta\sigma_1 + \zeta\sigma \|w^*\|_2}{2} + \frac{(\delta L_1 + \zeta L \|w^*\|_2 + \zeta\sigma)^2}{4\mu}$$

then the derivative of $n_e$ is:

$$\frac{\partial}{\partial n_e}\left(\frac{1}{2}A + \frac{1}{4\mu}B^2\right) = \frac{1}{2}\frac{\partial A}{\partial n_e} + \frac{B}{2\mu}\frac{\partial B}{\partial n_e}$$

and since both $\frac{\partial A}{\partial n_e}$ and $\frac{\partial B}{\partial n_e}$ are increasing functions and we know that:

$$\frac{\partial A}{\partial n_e}\bigg|_{n_e = 2\log_2\left(\frac{2(\ln 2)s\sigma_1 C}{\sigma \|w^*\|_2}\right)} = 0, \quad \frac{\partial B}{\partial n_e}\bigg|_{n_e = 2\log_2\left(\frac{2(\ln 2)sL_1 C}{L \|w^*\|_2 + \sigma}\right)} = 0$$

then we know the solution of $\frac{\partial}{\partial n_e}\left(\frac{1}{2}A + \frac{1}{4\mu}B^2\right) = 0$ is in the interval between $2\log_2\left(\frac{2(\ln 2)s\sigma_1 C}{\sigma \|w^*\|_2}\right)$ and $2\log_2\left(\frac{2(\ln 2)sL_1 C}{L \|w^*\|_2 + \sigma}\right)$.

$\square$

*Proof of Theorem 4.* In the denormal FPQ case, the set of quantization points are:

$$D = \left\{s \cdot \frac{x}{n_m} \cdot 2^{-\frac{n_e}{2}+3} \mid x = 0, 1, \cdots, n_m - 1\right\}$$

$$\cup \left\{s \cdot \left(1 + \frac{x}{n_m}\right) \cdot 2^y \mid x = 0, 1, \cdots, n_m - 1, \ y = -\frac{n_e}{2} + 3, \cdots, \frac{n_e}{2} - 1\right\}$$

then the parameters for the nonlinear quantization bound is:

$$\delta = s \cdot \frac{1}{n_m} \cdot 2^{-\frac{n_e}{2}+3} = \frac{8}{C} \cdot \frac{sn_e}{(\sqrt{2})^{n_e}}, \quad \zeta = \frac{1}{n_m}, \quad \eta = \frac{\zeta^2}{4(1+\zeta)} = \frac{1}{4n_m(n_m + 1)}$$

For NLQ-SGD, the noise ball size according to theorem 2 is:

$$\frac{\delta\sigma_1 + \zeta\sigma\,\|w^*\|_2}{2} + \frac{(\delta L_1 + \zeta L\,\|w^*\|_2 + \zeta\sigma)^2}{4\mu}$$

Denote this as $\frac{1}{2}A + \frac{1}{4\mu}B^2$. When $b$ is large, $\delta, \zeta, \eta$ are small and the dominating term for the noise ball is

$$A = \delta\sigma_1 + \zeta\sigma\,\|w^*\|_2 = \frac{8s\sigma_1}{C}\frac{n_e}{\left(\sqrt{2}\right)^{n_e}} + \sigma\,\|w^*\|_2\frac{1}{n_m} = \frac{8s\sigma_1}{C}\frac{n_e}{\left(\sqrt{2}\right)^{n_e}} + \sigma\,\|w^*\|_2\frac{n_e}{C}$$

let the derivative over $n_e$ to be 0 and we get:

$$\frac{\partial A}{\partial n_e} = \frac{8s\sigma_1}{C}\frac{1 - (\ln\sqrt{2})n_e}{\left(\sqrt{2}\right)^{n_e}} + \sigma\,\|w^*\|_2\frac{1}{C} = 0$$

denote $V(x) = x \cdot e^x$, and Lambert W function $W(y) = V^{-1}(y), y \geq -\frac{1}{e}$. then we need

$$\frac{\partial A}{\partial n_e} = \frac{8s\sigma_1}{C}\frac{1 - (\ln\sqrt{2})n_e}{\left(\sqrt{2}\right)^{n_e}} + \sigma\,\|w^*\|_2\frac{1}{C} = \frac{8s\sigma_1}{eC}V(1 - (\ln\sqrt{2})n_e) + \sigma\,\|w^*\|_2\frac{1}{C} = 0$$

thus we have:

$$n_e = 1 - \frac{2}{\ln 2}W\left(\frac{e\sigma\,\|w^*\|_2}{8s\sigma_1}\right), \quad b_e = \log_2\left[1 - \frac{2}{\ln 2}W\left(\frac{e\sigma\,\|w^*\|_2}{8s\sigma_1}\right)\right]$$

And when $b$ is small, $\delta, \zeta, \eta$ are large and the dominating term for the noise ball is

$$B = \delta L_1 + \zeta L\,\|w^*\|_2 + \zeta\sigma = \frac{8sL_1}{C}\cdot\frac{n_e}{\left(\sqrt{2}\right)^{n_e}} + (L\,\|w^*\|_2 + \sigma)\frac{n_e}{C}$$

let the derivative of $n_e$ to be 0 and we get:

$$\frac{\partial B}{\partial n_e} = \frac{8sL_1}{C}\frac{1 - (\ln\sqrt{2})n_e}{\left(\sqrt{2}\right)^{n_e}} + (L\,\|w^*\|_2 + \sigma)\frac{1}{C} = \frac{8sL_1}{eC}V(1 - (\ln\sqrt{2})n_e) + (L\,\|w^*\|_2 + \sigma)\frac{1}{C} = 0$$

thus we have:

$$n_e = 1 - \frac{2}{\ln 2}W\left(\frac{e(L\,\|w^*\|_2 + \sigma)}{8sL_1}\right) \quad b_e = 2\log_2\left[1 - \frac{2}{\ln 2}W\left(\frac{e(L\,\|w^*\|_2 + \sigma)}{8sL_1}\right)\right]$$

For $b$ such that neither the terms dominates the result, we know the noise ball size is:

$$\frac{1}{2}A + \frac{1}{4\mu}B^2 = \frac{\delta\sigma_1 + \zeta\sigma\,\|w^*\|_2}{2} + \frac{(\delta L_1 + \zeta L\,\|w^*\|_2 + \zeta\sigma)^2}{4\mu}$$

then the derivative of $n_e$ is:

$$\frac{\partial}{\partial n_e}\left(\frac{1}{2}A + \frac{1}{4\mu}B^2\right) = \frac{1}{2}\frac{\partial A}{\partial n_e} + \frac{B}{2\mu}\frac{\partial B}{\partial n_e}$$

and since both $\frac{\partial A}{\partial n_e}$ and $\frac{\partial B}{\partial n_e}$ are increasing functions and we know that:

$$\left.\frac{\partial A}{\partial n_e}\right|_{n_e = 1 - \frac{2}{\ln 2}W\left(\frac{e\sigma\,\|w^*\|_2}{8s\sigma_1}\right)} = 0, \quad \left.\frac{\partial B}{\partial n_e}\right|_{n_e = 1 - \frac{2}{\ln 2}W\left(\frac{e(L\,\|w^*\|_2 + \sigma)}{8sL_1}\right)} = 0$$

then we know the solution of $\frac{\partial}{\partial n_e}\left(\frac{1}{2}A + \frac{1}{4\mu}B^2\right) = 0$ is in the interval between $1 - \frac{2}{\ln 2}W\left(\frac{e\sigma\,\|w^*\|_2}{8s\sigma_1}\right)$ and $1 - \frac{2}{\ln 2}W\left(\frac{e(L\,\|w^*\|_2 + \sigma)}{8sL_1}\right)$.

$\square$

