# OpenReview forum: "Dimension-Free Bounds for Low-Precision Training"
_ICLR.cc/2019/Conference_

### Official Review · AnonReviewer3 · 2018-10-29
**Misleading title**

**Rating:** 6
**Confidence:** 4

**Review:**

This paper discusses conditions under which  the convergence of training models with low-precision weights do not rely on model dimension. Extensions to two kinds of non-linear quantization methods are also provided. The dimension-free bound of the this paper is achieved through a tighter bound on the variance of the quantized gradients.  Experiments are performed on synthetic sparse data and small-scale image classification dataset MNIST.

The paper is generally well-written and structure clearly. However, the bound for linear quantization is not fundamentally superior than previous bounds as the "dimension-free" bound in this paper is achieved by replacing the bound in other papers using l2 norm with l1 norm. Note that l1 norm is related to the l2 norm as: \|v\|_1 <= \sqrt{d}\|v\|_2, the bound can still be dependent on  dimension, thus the title may be misleading. Moreover, the assumptions  1 and 2 are much stronger than previous works, making the universality of the theory limited. The analysis on non-linear quantization is interesting, which can really theoretically improve the bound. It would be nice to see some more empirical results on substantial networks and  larger datasets which can better illustrate the efficacy of the proposed non-linear quantization.

Some minor issues:
1. What is HALP in the second contribution before Section 2?
2. What is LP-SVRG in Theorem 1?
3. What is \tilde{w} in Theorem 2?

---

> ### Author Response · Authors · 2018-11-26
> **Response to reviewer 3**
>
> We thank you for your careful reading and detailed review. Your review brings out the important parts of our work including the advantage of dimension-independence, the extension to nonlinear quantization, and the empirical validation.
>
> In our work, we use a different tight bound to show that the performance of low-precision training is not dependent on dimensionality inherently, but instead can be bound in terms of parameters such as \ell_1 and \sigma_1, which, in some cases, can be dependent on dimensionality but are not always. Thus under conditions where these parameters are fixed, we get a dimension-free result.
>
> We have two responses to your concerns.
>
> First, as shown in Fig 1(a), the standard dimension-dependent bound is in some sense tight, so we should expect to see classes of problems for which the performance depends strongly on $d$. For these classes of problems, our parameters L_1 and \sigma_1 will also increase strongly with $d$, as you point out. However, there are classes of problems for which this does not happen, and for the class we study in Figure 2(a), the performance does not depend on $d$ either, which is what our theory predicts. Our theory provides dimension-independent rates for low-precision SGD only when our assumptions (our bounds on the \ell_1 parameters) hold, not for all optimization problems in general.
>
> Second, even in the worst-case scenario when the parameters L_1 and \sigma_1 do depend strongly on dimension, our results in Table 1 show that, by using non-linear quantization, we can actually put those terms inside double \log and get a O(\log\log d) upper bound when it comes to the number of bits required. Although it can not be said to be fully dimension-independent, this is substantially better than the O(\log d) bound from previous work on linear quantization.
>
> Regarding the minor issues: LP-SVRG and HALP are two algorithms for low-precision training proposed in [De Sa et al., 2018], and we had extended our result to the analysis of these two algorithms, but we moved this part to the appendix due to the space constraint and caused this confusion. As our main contributions, as pointed out by the reviewers, do not depend on this analysis, we have decided to cut it from the appendix in our revised manuscript. This should help avoid confusion by allowing the paper to focus solely on the main-body claims about low-precision SGD. (And \tilde{w} in theorem 2 is actually a typo, which should be \bar w_T, same as what we wrote in theorem 1 as well as the theorems we added in the appendix.)
>
>
> Christopher De Sa, Megan Leszczynski, Jian Zhang, Alana Marzoev, Christopher R Aberger, Kunle Olukotun, and Christopher Ré. High-accuracy low-precision training. arXiv preprint arXiv:1803.03383, 2018.

---

### Official Review · AnonReviewer2 · 2018-11-05
**A solid contribution to understanding quantization for SGD**

**Rating:** 6
**Confidence:** 3

**Review:**

The paper considers the problem of low precision stochastic gradient descent. Specifically, they study updates of the form x_{t + 1} = Q (x_t - alpha * g_t), where g_t is a stochastic gradient, and Q is a quantization function. The goal is to produce quantization functions that simultaneously increase the convergence rate as little as possible, while also requiring few bits to represent. This is motivated by the desire to perform SGD on low precision machines.

The paper shows that under a set of somewhat nonstandard assumptions, previously studied quantization functions as well as other low precision training algorithms are able to match the performance of non-quantized SGD, specifically, losing no additional dimension factors. Previous papers, to the best of my knowledge, did not prove such bounds, except under strong sparsity conditions on the gradients. I did not check their proofs line-by-line however they seem correct at a high level.

I think the main discussion about the paper should be about the assumptions made in the analysis.  As the authors point out, besides the standard smoothness and variance conditions on the functions, some additional assumptions about the function must be made for such dimension independent bounds to hold. Therefore I believe the main contribution of this paper is to identify a set of conditions under which these sorts of bounds can be proven.

Specifically, I wish to highlight Assumption 2, namely, that the ell_1 smoothness of the gradients can be controlled by the ell_2 difference between the points, and Assumption 4, which states that each individual function (not just the overall average), has gradients with bounded ell_2 and ell_1 norm at the optimal point. I believe that Assumption 2 is a natural condition to consider, although it does already pose some limitations on the applicability of the analysis. I am less sold on Assumption 4; it is unclear how natural this bound is, or how necessary it is to the analysis.

The main pros of these assumptions are that they are quite natural conditions from a theoretical perspective (at least, Assumption 2 is). For instance, as the authors point out, this gives very good results for sparse updates. Given these assumptions, I don’t think it’s surprising that such bounds can be proven, although it appears somewhat nontrivial.  The main downside is that these assumptions are somewhat limiting, and don’t seem to be able to explain why quantization works well for neural network training. If I understand Figure 4b correctly, the bound is quite loose for even logistic regression on MNIST. However, despite this, I think formalizing these assumptions is a solid contribution.

The paper is generally well-written (at least the first 8 pages) but the supplementary material has various minor issues.

Smaller comments / questions:

- While I understand it is somewhat standard in optimization, I find the term “dimension-independent“ here somewhat misleading, as in many cases in practice (for instance, vanilla SGD on deep nets), the parameters L and kappa (not to mention L_1 and kappa_1) will grow with the dimension.

- Do these assumptions hold with good constants for training neural networks? I would be somewhat surprised if they did.

- Can one get dimension independent bounds for quantized gradients under these assumptions?

- The proofs after page 22 are all italicized.

- The brackets around expectations are too small in comparison to the rest of the expressions.

---

> ### Author Response · Authors · 2018-11-26
> **Response to reviewer 2**
>
> We thank you for a particularly detailed review and constructive feedback. Your review summarizes the highlights of our work and helps us understand what parts of our work are not explained well enough and may cause confusion.
>
> In our work, we use a different tight bound to show the dimension-independence of low-precision training under particular conditions, which are identified as Assumption 1-4. While Assumption 1 and 3 are standard assumptions on lipschitz continuity and strong convexity, we added assumption 2 and 4 to achieve a stronger result. Assumption 2 is analogous to Assumption 1 but used the \ell_1 norm instead of \ell_2. This is motivated by both theoretical analysis from lemma 4, i.e. the bound we showed in Fig 1(a), and empirical results where we observed the dimension-independence. And in Assumption 4, we bound the gradients at the optimum point both in \ell_1 and \ell_2 norm. The intuition for Assumption 4 is that, since the average gradient \nabla f = 1 / n \sum_i \nabla f_i is 0 at the optimum, the gradients samples \nabla f_i should not be too large and thus can be bounded by some constant \sigma and \sigma_1. This sort of assumption is actually necessary for the analysis of low-precision training, since otherwise we have no way to bound the variance of the gradient samples. For example, in a previous work on quantized net [Li et al., 2017], they assumed the bound on global gradient $G$, which is a stronger assumption than the \ell_2 part of our Assumption 4 since ours only requires the bound at the optimum point. These assumptions, though somewhat nonstandard due to their \ell_1 dependence, are natural to consider from a theoretical perspective, and are commonly observed in experiments, such as networks with sparse entries.
>
> In response to the smaller comments/questions, in order:
>
> 1. Our main contribution is identifying the conditions under which we can provide a convergence bound for low-precision training in which the dimension $d$ does not appear. We also introduced an analysis of non-linear quantization which strongly weakens the effect of the dimension term (put in a double \log) even without assuming any extra \ell_1 bounds. We used “dimension-independent” in this sense.
>
> 2. In some empirical settings, such as those with sparse entries, our assumptions do hold with good constants. Our assumptions do not hold for training neural networks, since that is a non-convex problem.
>
> 3. It seems very likely that we could prove dimension-independent bounds for methods using quantized gradients under the same assumptions. Basically the same analysis should work.
>
> 4 & 5. We have fixed the formatting errors, and we thank the reviewer for these detailed comments.
>
>
> Hao Li, Soham De, Zheng Xu, Christoph Studer, Hanan Samet, and Tom Goldstein. Training
> quantized nets: A deeper understanding. In Advances in Neural Information Processing Systems,
> pages 5813–5823, 2017.

---

### Official Review · AnonReviewer4 · 2018-11-17
**An in-depth study of quantization errors and quantized convex optimization in low-precision training**

**Rating:** 6
**Confidence:** 3

**Review:**

This paper provides an in-depth study of the quantization error in low-precision training and gives consequent bounds on the low-precision SGD (LP-SGD) algorithm for convex problems under various generic quantization schemes.

[pros]
This paper provides a lot of novel insights in low-precision training, for example, a convergence bound in terms of the L1 gradient Lipschitzness can potentially be better than its L2 counterpart (which is experimentally verified on specially designed problems).

I also liked the discussions about non-linear quantization, how they can give a convergence bound, and even how one could optimally choose the quantization parameters, or the number of {exponent, significance} bits in floating-point style quantization, in order to minimize the convergence bound.

The restriction to convex problems is fine for me, because otherwise essentially there is not a lot interesting things to say (for quantized problems it does not make sense to talk about “stationary points” as points are isolated.)

This paper is very well-written and I enjoyed reading it. The authors are very precise and unpretentious about their contributions and have insightful discussions throughout the entire paper.

[cons]
My main concern is that of the significance: while it is certainly of interest to minimize the quantization error with a given number of bits as the budget (and that’s very important for the deployment side), it is unclear if such a *loss-unaware* theory really helps explain the success of low-precision training in practice.

An alternative belief is that the success comes in a *loss-aware* fashion, that is, efficient feature extraction and supervised learning in general can be achieved by low-precision models, but the good quantization scheme comes in a way that depends on the particular problem which varies case by case. Admittedly, this is a more vague statement which may be harder to analyze or empirically study, but it sounds to me more reasonable for explaining successful low-precision training than the fact that we have certain tight bounds for quantized convex optimization.

[a technical question]
In the discussions following Theorem 2, the authors claim that the quantization parameters can be optimized to push the dependence on \sigma_1 into a log term -- this sounds a bit magical to me, because there is the assumption that \zeta < 1/\kappa, which restricts setting \zeta to be too large (and thus restricts the “acceleration” of strides from being too large) . I imagine the optimal bound only holds when the optimal choice of \zeta is indeed blow 1/\kappa?

---

> ### Author Response · Authors · 2018-11-27
> **Response to reviewer 4**
>
> We thank you for your positive feedback and pertinent questions.Your review summarizes the key points of our paper, including the study of LP-SGD bounds in terms of L1 gradient Lipschitz continuity, nonlinear quantization schemes and the assignment of exponent bits in floating-point quantization.
>
> Regarding the cons as you point out, we agree with you that our paper is based on a loss-unaware analysis, which may be limited. Our theory in this work gives us an idea of how dimension affects the performance of quantized SGD, by providing conditions under which its performance does not vary with dimension. Nevertheless, it is possible that a more loss-aware analysis could produce a tighter result (or even allow us to extend to a non-convex setting). Analyzing loss-aware explanations for the success of low-precision training in theory and in practice is a future direction of our work.
>
> Regarding the technical problem, when using logarithmic quantization, our theory for the convergence bound (Theorem 2) holds when we set \zeta to be less than 1 / \kappa, which is \mu / L. The big-O expression for the number of bits presented after Theorem 2 is an asymptotic analysis that assumes sufficiently small values of epsilon, for which the chosen value of \zeta satisfies \zeta < 1 / \kappa. Even without this simplifying asymptotic analysis, all the parameters that depend on \ell_1 are still within the double-logarithm. We revised the proof in B.3 to make this more clear and explicit.

---

### Author Response · Authors · 2018-11-27
**Revision updated**

Dear readers and reviewers, we have uploaded the revised version of our paper, and we made the following changes:

We fixed some typos and font problems.
We removed some confusing mentions of SVRG [1] and HALP [1] in the main body of the paper, which should have been moved to in the appendix before revision.
We had included an extension of our work to SVRG and HALP in the original appendix, but after feedback from the reviewers it seems that this part of our results caused confusion and did not add much to our claims. We’ve cut the extension to SVRG and HALP from the appendix so that our work is now focused entirely on the analysis of LP-SGD.


[1] Christopher De Sa, Megan Leszczynski, Jian Zhang, Alana Marzoev, Christopher R Aberger, Kunle Olukotun, and Christopher Ré. High-accuracy low-precision training. arXiv preprint arXiv:1803.03383, 2018.

---

### Meta-Review · Area_Chair1 · 2018-12-17

**Confidence:** 5
**Recommendation:** Reject

**Metareview:**

As the reviewers pointed out, the strength of the paper mostly comes from the analysis of the non-linear quantization which depends on the double log of the Lipschitz constants and other parameters. The AC and reviewers agree with the dimension-independent nature of the bounds, but also note that dimension-independent gound may not necessarily be significantly stronger than the dimension-dependent bounds as the metric of measuring the difficulty of the problem also matters. Although the paper does seem to lack result that shows the empirical benefit of the non-linear quantization. In considering the author response and reviewer comments, the AC decided that this comparison was indeed important for understanding the contribution in this work, and it is difficult to assess the scope of the contribution without such a comparison.